# Fair Allocation in Dynamic Mechanism Design

**Alireza Fallah**
University of California, Berkeley
afallah@berkeley.edu

**Michael I. Jordan**
University of California, Berkeley and INRIA, Paris
jordan@cs.berkeley.edu

**Annie Ulichney**
University of California, Berkeley
annie_ulichney@berkeley.edu

## Abstract

We consider a dynamic mechanism design problem where an auctioneer sells an indivisible good to two groups of buyers in every round, for a total of $T$ rounds. The auctioneer aims to maximize their discounted overall revenue while adhering to a fairness constraint that guarantees a minimum average allocation for each group. We begin by studying the static case ($T = 1$) and establish that the optimal mechanism involves two types of subsidization: one that increases the overall probability of allocation to all buyers, and another that favors the group which otherwise has a lower probability of winning the item. We then extend our results to the dynamic case by characterizing a set of recursive functions that determine the optimal allocation and payments in each round. Notably, our results establish that in the dynamic case, the seller, on the one hand, commits to a participation reward to incentivize truth-telling, and, on the other hand, charges an entry fee for every round. Moreover, the optimal allocation once more involves subsidization in favor of one group, where the extent of subsidization depends on the difference in future utilities for both the seller and buyers when allocating the item to one group versus the other. Finally, we present an approximation scheme to solve the recursive equations and determine an approximately optimal and fair allocation efficiently.

## 1 Introduction

Auctions play a pivotal role in allocation decisions across various domains, serving as a structured methodology for determining the distribution of resources based on bids. Housing, government contracts, electromagnetic spectrum rights, and advertising slots are a few of the many domains where allocations are determined by auctions. In many of the real-world applications of auction-based allocations, the resource in question is both valuable and limited, which raises concerns about fairness.

For instance, housing auctions are often governed by policies addressing the needs of low- and moderate-income households, such as Affordable Housing Quotas in the United Kingdom to tax credits and vouchers in the United States [20, 22]. Government procurement contracts for goods and services in the United States are subject to the United States Small Business Act which outlines target contracting rates across categories (e.g., woman-owned businesses, veteran-owned businesses, historically underutilized zones) [17]. The United States Federal Communications Commission drives initiatives ensuring , for instance, that small and regional companies maintain reasonable access to connectivity [35].

The importance of incorporating a consideration of fairness in important real-world use cases motivates recent work that extends classical work on auction design emphasizing economic efficiency to ensure outcomes do not disproportionately favor some bidders over others. Integrating fairness

38th Conference on Neural Information Processing Systems (NeurIPS 2024).

into auction designs is, however, a source of significant complexity, given that buyers often have different valuations, which can lead to strategic bidding behavior that may skew fairness.

In this paper, we focus on studying the question of fairness in a dynamic mechanism design setting, where a seller allocates an indivisible item through an auction at every round and for $T$ rounds to two groups of buyers, each group including $n$ buyers. At each round, the buyers' values are realized independently from an underlying distribution (which may differ between the two groups), and the buyers then submit their bids. The seller's goal is to decide on an allocation rule as a function of the submitted bids that maximizes their total discounted revenue with a discount factor $\delta$, while ensuring that each group receives a minimum allocation, or more specifically, that group $i$'s average discounted allocation is greater than or equal to some $\alpha_i$.

This fairness constraint enforces a proportional notion of fairness regarding the number of items each group wins. There are several advantages to adopting such a fairness notion. First, it is typically more straightforward from a policy perspective compared to, for instance, focusing on the total value won by each group. For example, most regulations concerning fair housing allocation emphasize the percentage of housing allocated to low-income groups as a measure of success [27]. Moreover, a buyer's value in private-value auctions is not solely determined by the intrinsic value of the item but also depends on the financial constraints of the buyers or asymmetric information about the item's value. Therefore, a proportional fairness constraint is a suitable choice in cases where one group's values are systematically lower than another's. For example, in the context of housing allocation, low-income individuals may assign lower monetary value to homes compared to wealthier individuals, as they are more concerned with basic shelter than with investment or resale value. In such scenarios, the allocation ratio is a more effective approach to mitigate allocation inequality compared to using buyers' value.

We start by considering the static case with $T = 1$. In the unconstrained case, it is well-known that the optimal allocation is the second-price auction with a reserve price, which allocates the item to the buyer with the highest bid, conditioned on the bid being above a certain reserve price. Here, we establish that the optimal allocation under the fairness constraint has two main differences: first, the optimal fair mechanism subsidizes one group that would otherwise not meet the target fairness constraint. Second, the optimal allocation may increase the chance of allocating the item to both groups in general by reducing the reserve price (again disproportionately in favor of one group).

We next focus on the general dynamic case. Here, we characterize the optimal allocation through a set of recursive functions. At any round $t$, we define the *residual minimum allocation*, which, roughly speaking, given the allocation so far, updates the fairness constraint as if we were to start at this round. We establish that, given the residual minimum allocation, we may have to allocate the item to one of the two groups to be on track to satisfy the fairness constraint overall, and that, in this case, we would simply allocate to the highest bidder in the group and charge them the second-highest bid within their group. However, in cases where we have the option to allocate to both groups, the design of the optimal allocation is more complicated because buyers in each group may consider underreporting their value in the current round to keep their chances of winning the item in future rounds higher, especially if they expect their value in future rounds to be higher.

In this case, we establish that, as the above reasoning suggests, there is a utility associated with not winning the item. This utility is the difference between a buyer's expected utility in future rounds if their group does not win the item this round, compared to the scenario where their group wins the item at the current round. We establish that, in the optimal allocation, and to incentivize buyers to report their values truthfully, the seller commits to paying them a participation reward if their group wins the item, and this reward is equal to the aforementioned net utility of not winning the item. On the other hand, the seller charges them an entry fee, which is equal to the expected reward payment that they would miss if they did not participate. Finally, we also prove that the optimal allocation rule is similar to the static case at a high level: when allocating an item to a group, it is assigned to the buyer with the highest bid. Second, the decision regarding which group receives the item takes the form of a subsidy. However, here, the amount of the subsidy depends on the above net utility of not winning the item and the seller's future utility from allocating the item to one group or the other.

As our results so far suggest, finding the optimal allocation requires solving a set of recursive functions, and as we see, the computational complexity could grow exponentially with the number of rounds. Our next contribution is to provide an efficient approximation scheme. In particular, we establish that, for any $\varepsilon > 0$, we can find an allocation that guarantees the same utility to the seller

and a fairness constraint of $(1 - \varepsilon)(\alpha_i - \varepsilon)$ for group $i$. This requires only $\mathcal{O}(\varepsilon^{-1/(1-\delta)})$ calls to an oracle that computes integrals. We further provide a constant approximation scheme which requires $\text{Poly}(1/(1 - \delta))$ calls to the oracle, which is more applicable to scenarios where $\delta$ is close to one. We conclude our paper by providing a simple numerical experiment to illustrate the impact of the fairness constraint on the utility of both the seller and users.

**Related work:** Our work is related to the growing literature on questions related to fairness in mechanism design [15, 36, 24]. Many works in fair mechanism design with an algorithmic focus draw from notions of fairness central to computer science and machine learning [19]. For instance, a prominent related area of fair algorithmic mechanism design studies frameworks that ensure equity in the distribution of online advertisements across protected user attributes in the setting of online auctions, e.g. [14], [13], and [21].

Our work is also related to a rich literature on fair division that studies fair allocation without considering the strategic behavior of agents, e.g. [8], [16], [11], [34], [12], [3]. Our work joins the set of works that consider both fairness and incentive compatibility in allocation. Where [25], [9], [12] consider truthful fair allocations with respect to variants of envy-freeness and [7], [6] consider more broadly the feasibility of different notions of fairness in a strategic setting, we join works that adopt a proportional notion of fairness, e.g, [1], [2], [4], [15]. We further depart from these works in our focus on revenue optimization (instead of, e.g., optimizing social welfare [36]) and in our treatment of the dynamic strategic environment.

Another group of related works studies the impact of budgetary and liquidity constraints on auction revenue, e.g. [26], and [18]. These works' motivation for considering budget constraints aligns with the setting where fair allocation guarantees are relevant. In particular, both settings drop the assumption that a bidder's valuation is accurately reflected by their willingness to pay. The setting of asymmetric budgetary constraints studied by [32] motivates the relevance of our work's introduction of guarantees for relative allocative fairness between groups. We build upon this set of work by introducing a framework where the allocative limitations of both homogeneous and heterogeneous budget constraints can be simultaneously overcome as efficiently as possible. A separate set of related works empirically studies the impact on efficiency of subsidizing targeted groups of bidders in procurement auctions [23], [5]. These works complement ours by pointing to bid subsidies as the cost-optimal mechanism to enforce fairness.

The closest setting to ours is that of [31] which examines a similar notion of fairness in the single-round (static) case, but only for one group. We depart from their work as we consider the dynamic case and present an approximation scheme. Additionally, even in the static case, we consider a minimum allocation constraint for both groups, which introduces a second type of subsidization.

## 2  Model

We consider a monopolist seller that interacts with two groups of buyers over $T$ rounds,[1] with $n$ buyers in each group.[2] At each round $t \in [T]$,[3] the seller conducts an auction to sell a single indivisible item. The private value of buyer $k \in [n]$ from group $i \in [2]$ (or simply, buyer $(i, k)$) for this item is denoted by $v_{i,k}^t$, which is drawn from the publicly known and full support continuous distribution with density $f_i^t : [\underline{v}_i^t, \bar{v}_i^t] \to \mathbb{R}$. We also denote its cumulative distribution function by $F_i^t(\cdot)$. We assume the buyers' values are independent.

**Timeline of the auction:**  At each time $t$, buyers' private values are realized, and then each buyer $(i, k)$ submits a bid $b_{i,k}^t$. We denote the vector of bids at round $t$ by $\boldsymbol{b}^t$. The seller then conducts the auction and decides on the allocation $\boldsymbol{x}^t(\boldsymbol{b}^t, \boldsymbol{h}^t)$ where $\boldsymbol{h}^t$ denotes the public history of allocations up to (but not including) time $t$, i.e., which buyers received the items in the previous $t - 1$ rounds. In particular, $x_{i,k}^t(\boldsymbol{b}^t, \boldsymbol{h}^t) \in \{0, 1\}$ denotes whether the $k$th buyer in group $i$ gets the item in round $t$. Notice that since we allocate a single item at each round, we should have $\sum_{i=1}^2 \sum_{k=1}^n x_{i,k}^t(\boldsymbol{b}^t, \boldsymbol{h}^t) \leq 1$ for every $t \in [T]$. Finally, buyer $(i, k)$ makes the payment $p_{i,k}^t(\boldsymbol{b}^t, \boldsymbol{h}^t)$

---

[1]We demonstrate that the results extend to the case of multiple groups of buyers in an extended version available at https://arxiv.org/abs/2406.00147.

[2]Here, for simplicity in notation, we assume that the two groups are of equal size $n$. However, our analysis and results apply to the general case, including unequal group sizes and scenarios with more than two groups.

[3]For any integer $N$, $[N]$ denotes the set $\{1, \ldots, N\}$.

to the seller. The vector of payments at time $t$ is denoted by $\boldsymbol{p}^t(\boldsymbol{b}^t, \boldsymbol{h}^t)$. Notice that the mechanism is determined by the allocation and payment functions $(\boldsymbol{x}^{1:T}, \boldsymbol{p}^{1:T})$.

**Utility functions:**  The utility function of buyer $(i, k)$ at time $t$ is given by $\mathcal{U}_{i,k}^t(v_{i,k}^t; \boldsymbol{b}^t, \boldsymbol{h}^t) := x_{i,k}^t(\boldsymbol{b}^t, \boldsymbol{h}^t)v_{i,k}^t - p_{i,k}^t(\boldsymbol{b}^t, \boldsymbol{h}^t)$. Buyer $(i, k)$ and seller's overall utilities are, respectively, given by:

$$\mathcal{U}_{i,k}(\boldsymbol{v}_{i,k}^{1:T}; \boldsymbol{b}^{1:T}, \boldsymbol{h}^{1:T}) := \sum_{t=1}^{T} \delta^{t-1} \mathcal{U}_{i,k}^t(v_{i,k}^t; \boldsymbol{b}^t, \boldsymbol{h}^t) \text{ and } \mathcal{U}_0(\boldsymbol{b}^{1:T}, \boldsymbol{h}^{1:T}) := \sum_{t=1}^{T} \delta^{t-1} \sum_{(i,k)} p_{i,k}^t(\boldsymbol{b}^t, \boldsymbol{h}^t) \quad (1)$$

where $\delta \in (0, 1]$ is the discount factor. Our goal is to identify mechanisms that maximize the seller's expected utility, subject to certain fairness constraints ensuring a minimum allocation is guaranteed to each group. We next formally define these fairness constraints.

**The fairness constraint:**  The seller, acting as the auctioneer, aims to ensure that a minimum allocation is guaranteed to each group in this dynamic setting. More specifically, let $\alpha_i$ represent the minimum (discounted) average allocation promised to group $i$, with the condition that $\alpha_1 + \alpha_2 \leq 1$. Then, at each round $t$, the seller ensures that the discounted average of the items allocated to group $i$ thus far, combined with the expected allocation in the remaining rounds, is at least $\alpha_i$:

$$\frac{1-\delta}{1-\delta^T} \left( \sum_{\tau=1}^{t-1} \delta^{\tau-1} \sum_{k=1}^{n} x_{i,k}^\tau + \mathbb{E}\left[ \sum_{\tau=t}^{T} \delta^{\tau-1} \sum_{k=1}^{n} x_{i,k}^\tau \right] \right) \geq \alpha_i, \quad (2)$$

where the expectation is taken over the realization of the private values at time $t$ and the future rounds. We call the auction *infeasible* at round $t$ if it is not possible to satisfy the fairness constraint over the remaining rounds.

Notice that this condition is ex ante with respect to the allocation at time $t$ onwards as we find this condition more compatible compared to an ex post fairness in a setting with indivisible goods. Also, it is straightforward to verify that if the condition (2) holds for $t = T$, then it would hold for any $t \leq T$. However, as we will elaborate later, this condition determines the set of feasible allocations for any $t \leq T - 1$. For instance, suppose $T = 4$ and $\alpha_1 = \alpha_2 = 1/3$. Then, if we allocate the item to group 1 in the first three rounds, we cannot satisfy condition (2) for group 2 in the last round.

**Direct truthful mechanisms:**  The dynamic revelation principle states that, without loss of generality, we can focus on direct, truthful mechanisms where buyers bidding truthfully is a Nash equilibrium [29]. It is important to note that the dynamic revelation principle requires truthful reporting only on the equilibrium path; that is, assuming all buyers have been truthful in the past. However, in our setting, this would imply truthful reporting under any history, even if a buyer has previously been dishonest (and we are off the equilibrium path). To see why this is the case, note that the utility of each buyer depends on their current (private) value and all past bids (but not on the past true values). Thus, when it is optimal for the buyer to bid truthfully when past reports have been truthful, it remains optimal for them to bid truthfully even if some buyers have lied in the past. This argument generally holds true for Markovian environments (see [33, 37] for a detailed discussion on this topic.)

In this paper, we use the *periodic ex post incentive compatibility* (see [10] for a discussion on this definition and its comparison with the weaker notion of Bayesian incentive compatibility in dynamic mechanism design). More specifically, for any mechanism $(\boldsymbol{x}^{1:T}, \boldsymbol{p}^{1:T})$, the following Markovian decision problem finds the best bid for buyer $k$ from group $i$, assuming that all other buyers are reporting their values truthfully:

$$U_{i,k}^t(\boldsymbol{v}^t, \boldsymbol{h}^t) := \max_{b_{i,k}^t} \left\{ \mathcal{U}_{i,k}^t \left( v_{i,k}^t; b_{i,k}^t, \boldsymbol{v}_{-(i,k)}^t, \boldsymbol{h}^t \right) + \delta \, \mathbb{E}\left[ U_{i,k}^{t+1}(\boldsymbol{v}^{t+1}, \boldsymbol{h}^{t+1}) \Big| \boldsymbol{h}^t \right] \right\}, \quad (3)$$

with the convention that $U_{i,k}^{T+1}(\cdot, \cdot) := 0$. This is a recursive formula in which the buyer identifies the optimal bid by maximizing the sum of their utility at the current moment and the maximum of what they could achieve in the future, while they assume that all other buyers are bidding truthfully. A mechanism is called ex post incentive compatible if the maximum is achieved through truthful bidding. The formal definition is provided below.

**Definition 1.** *A mechanism $(\boldsymbol{x}^{1:T}, \boldsymbol{p}^{1:T})$ is ex post incentive compatible (EPIC) if:*

$$v_{i,k}^t \in \arg\max_{b_{i,k}^t} \left\{ \mathcal{U}_{i,k}^t \left( v_{i,k}^t; b_{i,k}^t, \boldsymbol{v}_{-(i,k)}^t, \boldsymbol{h}^t \right) + \delta \, \mathbb{E}\left[ U_{i,k}^{t+1}(\boldsymbol{v}^{t+1}, \boldsymbol{h}^{t+1}) \Big| \boldsymbol{h}^t \right] \right\} \forall i, k, t, \boldsymbol{v}^t, \boldsymbol{h}^t. \quad (4)$$

Note that the EPIC mechanism can be identified through backward induction, starting at the last round and progressing back to the first. By doing so, at time $t$, we would realize that maximum utility in future rounds is achieved through truthful bidding. Consequently, the term $U_{i,k}^{t+1}(\boldsymbol{v}^{t+1}, \boldsymbol{h}^{t+1})$ could be replaced with the utility assuming that everyone reports truthfully. In other words, to identify the EPIC mechanism, it is sufficient to find a mechanism that satisfies the following condition:

$$v_{i,k}^t \in \arg\max_{b_{i,k}^t} \left\{ \mathcal{U}_{i,k}^t \left( v_{i,k}^t; b_{i,k}^t, \boldsymbol{v}_{-(i,k)}^t, \boldsymbol{h}^t \right) + \delta \, \mathbb{E} \left[ \sum_{\tau=t+1}^T \mathcal{U}_{i,k}^\tau (v_{i,k}^\tau; \boldsymbol{v}^\tau, \boldsymbol{h}^\tau) \Big| \boldsymbol{h}^t \right] \right\}. \tag{5}$$

Finally, we state the individual rationality assumption or the participation condition. This assumption ensures that, at each round, each buyer (knowing only their value) would weakly prefer to participate in the auction. In other words, each buyer's expected utility is weakly higher if they participate rather than choosing their outside option, which is skipping that round.

**Definition 2.** *A mechanism $(\boldsymbol{x}^{1:T}, \boldsymbol{p}^{1:T})$ is individually rational (IR) if for all $i, k, t, v_{i,k}^t$, and $\boldsymbol{h}^t$:*

$$\mathbb{E} \left[ U_{i,k}^t (\boldsymbol{v}^t, \boldsymbol{h}^t) \right] \geq \delta \, \mathbb{E} \left[ U_{i,k}^{t+1}(\boldsymbol{v}^{t+1}, (\boldsymbol{h}')^{t+1}) \Big| \boldsymbol{h}^t \right], \text{`} \tag{6}$$

*where $(\boldsymbol{h}')^{t+1}$ denotes the history under which buyer $(i, k)$ has not participated at round $t$, and the expectations are taken over $\boldsymbol{v}_{-(i,k)}^t$ and $\boldsymbol{v}^{t+1:T}$.[4]*

The left hand side of (6) demonstrates the expected utility of buyer $(i, k)$ at time $t$ and thereafter, assuming their participation in round $t$. However, the right hand side of (6) shows the expected utility of buyer $(i, k)$ when they skip the $t$-th round (and receive no utility in the $t$-th round).

Throughout our analysis, we adopt the following regularity assumption on the distribution of values that is common in mechanism design literature. This assumption applies to a variety of distributions, including uniform, exponential, and normal distributions.[5]

**Assumption 1.** *For any $i$ and $t$, the distribution $F_i^t$ is regular, i.e., the virtual value function $\phi_i^t(v) := v - \frac{1 - F_i^t(v)}{f_i^t(v)}$ is increasing in $v$.*

For the rest of the paper, our goal is to find an EPIC and IR mechanism $(\boldsymbol{x}^{1:T}, \boldsymbol{p}^{1:T})$ that maximizes the expected seller utility subject to the fairness constraint (2). We start our analysis by discussing the static case, i.e., $T = 1$, which will be used in our analysis for the dynamic setting.

## 3 The static case

To simplify the notation, we drop the dependence of the parameters on time $t$ in this section. Here, our fairness constraint (2) reduces to $\mathbb{E} \left[ \sum_{k=1}^n x_{i,k} \right] \geq \alpha_i$, where the expectation is taken over a single realization of the private values.

It is well-known that in the absence of the fairness constraint, and under Assumption 1, the item is allocated to the buyer who has the highest virtual value, provided that at least one buyer's virtual value is non-negative. Conversely, the seller does not allocate the item if all virtual values are negative. This mechanism is known as the second-price auction (or Vickrey auction) with reserve pricing, as detailed in [30]. This allocation is depicted in Figure 1a. We next present the main result of this section which establishes the optimal allocation under the fairness constraint.

**Theorem 1.** *Suppose Assumption 1 holds. For any $i \in [2]$, let $v_i := \max_k v_{i,k}$ be the maximum value among buyers in group $i$. Then, the following results hold for the optimal allocation:*

*(i) If the item is allocated to group $i$, then it is allocated to the buyer in group $i$ with the highest value, i.e., if $x_{i,k} = 1$, then $v_{i,k} = v_i$.*

---

[4]In the case of $n = 1$, a buyer's non-participation could lead to the infeasibility of the auction as the item cannot be allocated to their group in that round. To avoid such special cases, in the case of $n = 1$, we assume that we may still allocate the item to their group even if they do not participate, but this potential allocation does not count towards their outside option utility.

[5]We demonstrate that the results in both the static and dynamic cases hold when this is assumption is weakened via ironing in an extended version available at https://arxiv.org/abs/2406.00147.

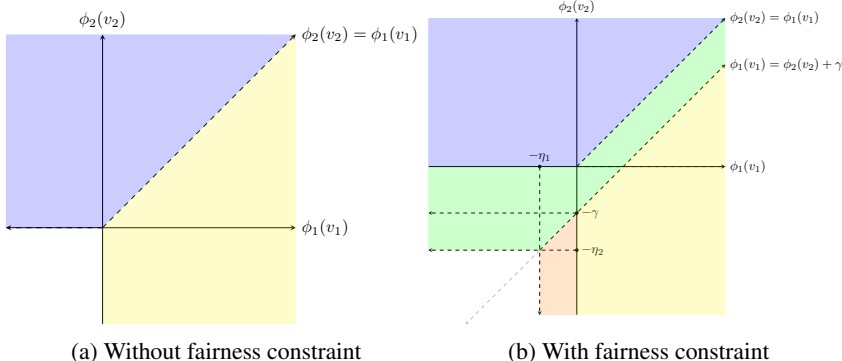

(a) Without fairness constraint         (b) With fairness constraint

Figure 1: Optimal allocation in the static case.

*(ii) The allocation decision at the group level depends only on the maximum value of the two groups, $v_1$ and $v_2$. For any $i$, let $G_i$ denote the pair $(v_1, v_2)$ for which the item is allocated to group $i$. Then, there exists $\eta_1, \eta_2 \geq 0$ and $\gamma$ with $\eta_2 = \eta_1 + \gamma$ such that (up to a measure-zero set)*

$$G_1 = \{(v_1, v_2) \mid \phi_1(v_1) \geq \phi_2(v_2) + \gamma \text{ and } \phi_1(v_1) \geq -\eta_1\}, \tag{7a}$$

$$G_2 = \{(v_1, v_2) \mid \phi_2(v_2) \geq \phi_1(v_1) - \gamma \text{ and } \phi_2(v_2) \geq -\eta_2\}. \tag{7b}$$

The proof of this result (along with all other proofs) can be found in the appendix. An example of the optimal allocation rule with $\gamma > 0$ is depicted in Figure 1b where group one is allocated the item in the yellow and orange regions and group two is allocated the item in the blue and green regions.

The theorem follows from the fact that the loss in the seller's utility increases in Euclidean distance from the boundary of the unconstrained optimal allocation $\phi_1(v_1) = \phi_2(v_2)$. Therefore, virtual valuations of the under-allocated group that are closest to those of the over-allocated group, specifically within a distance $\gamma$, are the set that minimize the cost of reallocating to the target group to achieve the desired balance between groups. In the case of insufficient unconstrained allocation, the further modification of allocating in cases where the item remains unallocated in the unconstrained mechanism is necessary.

As Theorem 1 shows, the optimal fair mechanism subsidizes one group that would otherwise not meet target allocation levels. To maximize revenue while satisfying a given fairness constraint, we deviate from the unconstrained allocation in the most cost efficient way, that is, we reassign the good to a lower valuation bidder in cases that least impact expected seller revenue.

Also, notice that, when there is no fairness constraint, the seller does not allocate the item if the maximum bid is below their *reserve price* $r = \min\{\phi_1^{-1}(0), \phi_2^{-1}(0)\}$. Thus, it is possible that, in the unconstrained optimal mechanism, the total probability of allocation across both groups be less than the sum of the target allocation levels $\alpha_i$. This is the intuition behind why $\eta_1, \eta_2$ may both be greater than zero in some cases. In other words, we must simultaneously enforce that the item is allocated more often in general and that the relative rates of allocation between groups is sufficiently balanced.

We conclude this section with a result which helps us to identify $\gamma$, $\eta_1$, and $\eta_2$.

**Proposition 1.** *Let $\gamma$, $\eta_1$, and $\eta_2$ correspond to the optimal allocation given by Theorem 1. Then, if $\gamma > 0$, i.e., the seller strictly subsidizes in favor of the second group, we have $\mathbb{P}(G_2) = \alpha_2$. Moreover, if $\eta_1 > 0$, we have $\mathbb{P}(G_1) = \alpha_1$. On the other hand, if $\gamma < 0$, i.e., the seller strictly subsidizes in favor of the first group, we have $\mathbb{P}(G_1) = \alpha_1$. Moreover, if $\eta_2 > 0$, we have $\mathbb{P}(G_2) = \alpha_2$.*

The probability $\mathbb{P}(\cdot)$ above is defined with respect to the distribution of the pair of maximum values $(v_1, v_2)$ with CDF $F_{\max}(v_1, v_2) = F_1(v_1)^n F_2(v_2)^n$. The intuition behind this proposition is that the seller satisfies the fairness constraint while maximizing their utility by deviating from the optimal unconstrained only as much as is necessary to satisfy the fairness constraint with equality.

# 4 The dynamic case

We next turn our attention to the dynamic case. Here, our main approach involves using backward induction. The analysis from the previous section serves as the basis for the optimal allocation at the last round, i.e., $t = T$. In this section, we describe how we proceed backward to find the optimal dynamic auction under the fairness constraint. We make the following assumption which simplifies the characterization of the optimal mechanism and allows us to focus on the insights. In the appendix, we discuss how our analysis extends to cases where this assumption does not hold.

**Assumption 2.** *We assume the seller must allocate the item in any round, except possibly the last.*

We start our analysis by making a number of definitions and observations. Suppose we are at the beginning of round $t$. Given (2), our allocation at round $t$ and the future rounds should satisfy the following constraint:

$$\sum_{\tau=t}^{T-1} \delta^{\tau-t} \sum_{k=1}^{n} x_{i,k}^{\tau} + \delta^{T-t} \mathbb{E}\left[\sum_{k=1}^{n} x_{i,k}^{T}\right] \geq R_i^t := \frac{1}{\delta^{t-1}}\left(\frac{1-\delta^T}{1-\delta}\alpha_i - \sum_{\tau=1}^{t-1} \delta^{\tau-1} \sum_{k=1}^{n} x_{i,k}^{\tau}\right), \quad (8)$$

We refer to the right-hand side as the *residual minimum allocation* for group $i$. It is noteworthy that equation (8) can be interpreted as if the auction is starting at time $t$, with the fairness constraint given by $R_i^t(1-\delta)/(1-\delta^{T-t})$. We next make the following simple observation:

**Fact 1.** *Given the allocation at time $t$, the residual minimum allocation at time $t + 1$, i.e., $R_i^{t+1}$, is equal to $\frac{1}{\delta}(R_i^t - 1)$ when the item is allocated to group $i$ and equal to $\frac{1}{\delta}R_i^t$ otherwise.*

We next introduce two interim functions. For the sake of these two definitions, assume the auction starts at time $t$, i.e., the utility at any time $\tau$, for $\tau \geq t$, is discounted by a factor of $\delta^{\tau-t}$. With this assumption, and for a given pair of residual minimum allocations $(R_1^t, R_2^t)$, we define $\mu^t(R_1^t, R_2^t)$ as the (discounted) sum of the expected utility of the seller at time $t$ and thereafter, under the optimal mechanism satisfying equation (8), prior to the realization of values. Similarly, $\nu_i^t(R_1^t, R_2^t)$ represents the expected utility of a buyer in group $i$. It is worth noting that, given the distribution of values within each group is similar, $\nu_i^t$ does not depend on $k$ (the buyer's index within the group). We also set the these functions equal to $-\infty$ if there is no feasible auction satisfying (8). The following result regarding $\nu_i^T(\cdot, \cdot)$ and $\mu^T(\cdot, \cdot)$ is a corollary of Theorem 1. Notice that the last round can be seen as a static auction with fairness constraint for group $i$ given by $R_i^T$.

**Corollary 1.** *At round $T$, and for a given pair of residual minimum allocations $(R_1^T, R_2^T)$, we have*

$$\nu_i^T(R_1^T, R_2^T) = \frac{1}{n}\int_{G_i} \frac{1-F_i^T(v_i)}{f_i^T(v_i)} dF_{max}^T(v_1, v_2), \quad (9a)$$

$$\mu^T(R_1^T, R_2^T) = \int_{G_1} \phi_1^T(v_1)dF_{max}^T(v_1, v_2) + \int_{G_2} \phi_2^T(v_2)dF_{max}^T(v_1, v_2), \quad (9b)$$

*where $F_{max}^T(v_1, v_2) = F_1^T(v_1)^n F_2^T(v_2)^n$ and $G_i$'s are defined in Theorem 1.*

This corollary is proved in the appendix. Now, suppose we are in the $t$-th round, conducting the backward induction. Consequently, we have access to $\nu_i^{t+1}(\cdot, \cdot)$ (for $i \in [2]$) and $\mu^{t+1}(\cdot, \cdot)$. Next, we will establish the optimal allocation for round $t$ and describe how to compute $\nu_i^t(\cdot, \cdot)$ and $\mu^t(\cdot, \cdot)$ accordingly. We have three main regimes:

**(I) When the auction is infeasible:** Suppose that no matter whether the seller gives the item to the first or second group, they would not be able to satisfy the residual minimum allocation constraint in the remaining rounds. In this case, we declare the auction is infeasible at round $t$.

**Fact 2.** *For a given pair of residual minimum allocations $(R_1^t, R_2^t)$, suppose we have $\mu^{t+1}((R_1^t - 1)/\delta, R_2^t/\delta)$ and $\mu^{t+1}(R_1^t/\delta, (R_2^t - 1)/\delta) = -\infty$. Then, there is no feasible auction satisfying the residual minimum allocation constraints, and hence, we set $\mu^t(R_1^t, R_2^t)$ and $\nu_i^t(R_1^t, R_2^t)$, for both $i \in [2]$, equal to $-\infty$.*

To better highlight this proposition, let us consider a simple example. Suppose $T = 2$, $\delta = 0.5$, and $\alpha_1 = \alpha_2 = 0.4$. In this scenario, $R_1^1 = R_2^1 = 0.6$. Now, the residual minimum allocation for the second round for the group that doesn't receive the item in the first round would be 1.2, which is not feasible. Hence, there is no feasible auction that satisfies the fairness constraint.

**(II) When the item must be allocated to a certain group:** Suppose, for instance, that if the seller allocated the item to the second group, they would not be able to satisfy the resulting residual minimum allocation constraints in the remaining rounds. However, allocating the item to the first group would result in a feasible auction. Therefore, in this case, the seller must allocate the item (if keeping it were feasible, then allocating it to the second group would also have been feasible), and moreover, the item must be allocated to the first group. The following result formalizes the optimal allocation for this scenario.

**Proposition 2.** *Suppose Assumption 1 holds. For a given pair of residual minimum allocations* $(R_1^t, R_2^t)$ *and some* $i \in [2]$, *suppose we have* $\mu^{t+1}(R_i^t/\delta, (R_{-i}^t - 1)/\delta) = -\infty$ *but* $\mu^{t+1}((R_i^t - 1)/\delta, R_{-i}^t/\delta) > -\infty$. *Then, the seller allocates the item to the buyer with the highest value in group* $i$, *with the payment being equal to the second-highest value within the same group.*

Intuitively, if the item must be allocated to the first group, the question of who within that group receives the item essentially boils down to a static Vickrey auction: the buyer offering the highest value acquires the item and pays the price equal to the second highest value in the group. It is important to note that, in this scenario, there is no reserve price (and we do not need to explicitly impose Assumption 2) because the seller is obliged to allocate the item to meet the fairness constraint. Lastly, the update of interim functions are provided in Appendix A.4.1.

**(III) When allocation to both groups is feasible:** Suppose that whether the seller allocates the item to the first or second group, they still have the opportunity to meet the allocation constraint in future rounds. Now, what should be the seller's allocation strategy to maximize their utility?

Note that the expected utility of buyers in each group changes based on whether or not they receive the item in the current round. The difference between these two scenarios can be interpreted as the expected utility of not receiving the item. More formally, for a given pair of residual minimum allocations $(R_1^t, R_2^t)$, and when allocation to both groups is feasible, the net utility of not receiving the item for group $i$ is defined as

$$\Delta_i^t(R_1^t, R_2^t) := \nu_i^{t+1}(R_i^t/\delta, (R_{-i}^t - 1)/\delta) - \nu_i^{t+1}((R_i^t - 1)/\delta, R_{-i}^t/\delta). \quad (10)$$

It is worth noting that, interestingly, the net utility of not receiving the item could be negative in some case (See Appendix A.5). Lastly, we define $\Delta_0^t(R_1^t, R_2^t)$ as the difference in seller's future utility by allocating the item to the first group, compared to allocating it to the second group, i.e.,

$$\Delta_0^t(R_1^t, R_2^t) := \mu^{t+1}((R_1^t - 1)/\delta, R_2^t/\delta) - \mu^{t+1}((R_2^t - 1)/\delta, R_1^t/\delta). \quad (11)$$

**Theorem 2.** *Suppose Assumptions 1 and 2 hold. For a given pair of residual minimum allocations* $(R_1^t, R_2^t)$, *suppose both* $\mu^{t+1}(R_1^t/\delta, (R_2^t - 1)/\delta)$ *and* $\mu^{t+1}((R_1^t - 1)/\delta, R_2^t/\delta)$ *are finite. Then, the seller allocates the item to the buyer with the highest value in group* $i$ *if* $(v_1^t, v_2^t) \in G_i^t$, *where* $v_j^t := \max_k v_{j,k}^t$ *is the maximum value among buyers in group* $j$ *and* $G_i^t$ *is given by*

$$G_i^t := \left\{ (v_1, v_2) \,\middle|\, \phi_i^t(v_i) - \phi_{-i}^t(v_{-i}) \geq n\delta \left( \Delta_i^t(R_1^t, R_2^t) - \Delta_{-i}^t(R_1^t, R_2^t) \right) + (-1)^i \delta \Delta_0^t(R_1^t, R_2^t) \right\}.$$

*Moreover, let* $i^*$ *denote the index of the winner group. Then, the payment of buyer* $(i, k)$ *is given by*

$$v_{i,k}^t x_{i,k}^t(v_{i,k}^t, \boldsymbol{v}_{-(i,k)}^t) - \int_{\underline{v}_i^t}^{v_{i,k}^t} x_{i,k}^t(z, \boldsymbol{v}_{-(i,k)}^t) dz + \delta \Delta_i^t(R_1^t, R_2^t)(\zeta_i^t - \mathbb{1}(i^* = i)), \quad (12)$$

*where* $\zeta_i^t$ *denotes the probability of group* $i$ *winning the item if the auction runs with* $n - 1$ *buyers from their group instead of* $n$ *buyers.*

The explicit characterization of $\zeta_i^t$'s along with the update of interim functions are provided in Appendix A.6. It is worth emphasizing that while the term $\zeta_i^t$ depends on the allocation rule of an auction with $2n - 1$ buyers, we can derive it in a non-recursive manner. The reason is that finding the allocation, similar to $G_i^t$ above, does not require running the smaller-size auction. Next, we would like to draw a few insights from this result. First, notice that, similar to the static case, the boundary of allocation rule is a linear function of the maximum virtual values of both group.

Next, note that the payment function consists of three terms. The first term, $v_{i,k}^t x_{i,k}^t(v_{i,k}^t, \boldsymbol{v}_{-(i,k)}^t) - \int_{\underline{v}_i^t}^{v_{i,k}^t} x_{i,k}^t(z, \boldsymbol{v}_{-(i,k)}^t) dz$, represents what only the winner pays and is equal to the minimum bid

they could have made to still win the item. This is indeed similar to the second-price auction in which the winner's payment is equal to the second highest bid. To better understand the other two terms, and for the sake of discussion, suppose $\Delta_i^t(R_1^t, R_2^t)$'s are non-negative, indicating a non-negative net future utility associated with not receiving the item at the current time (because it means your chance of receiving it in the future increases). The second term of the payment function is $-\delta\Delta_i^t(R_1^t, R_2^t)\mathbb{1}(i^* = i)$, which is negative and thus represents a transfer from the seller to the buyers (we call this *participation reward*). Only the buyers from the group that wins the item receive this payment. To understand why such a reward is needed from the seller, notice that when a buyer's group wins the item, they are in fact losing an amount of $\delta\Delta_i^t(R_1^t, R_2^t)$ in their future utility because their group has a lower chance of winning in the future. Hence, in some sense, this reduces the value of the item in the current round, and the buyers may find it profitable to underreport their value. That said, this payment serves to incentivize truthful reporting. Finally, the last term in the payment function is $\delta\Delta_i^t(R_1^t, R_2^t)\zeta_i^t$, which is a payment that every buyer should make to the seller. The rationale here is that if the buyer skips the current round, then if their group wins, which happens with probability $\zeta_i^t$, their group would receive a payment equal to $\delta\Delta_i^t(R_1^t, R_2^t)$ from the seller, as we elaborated above. Hence, the seller charges the same amount as the *entry fee*.

Lastly, notice that in the allocation rule $G_i^t$, there is a threshold of $n\delta\left(\Delta_i^t(R_1^t, R_2^t) - \Delta_{-i}^t(R_1^t, R_2^t)\right) + (-1)^i\delta\Delta_0^t(R_1^t, R_2^t)$ which determines which group receives the item. Notice that this threshold increases as $\Delta_i^t(R_1^t, R_2^t)$ increases, meaning that the seller is less willing to give the item to group $i$. This is because a higher $\Delta_i^t(R_1^t, R_2^t)$ means that the seller has to make a higher payment to group $i$'s buyers if they win. Additionally, this threshold decreases for the first group when $\Delta_0^t(R_1^t, R_2^t)$ increases, indicating that the seller is more willing to allocate the item to the first group when $\Delta_0^t(R_1^t, R_2^t)$ is higher. This is intuitive as this term represents the seller's future net utility by allocating the item to the first group compared to allocating it to the second group.

**An approximation scheme:** Our results so far offer an exact characterization of the optimal dynamic allocation. However, due to the recursive nature of the analysis, the computational complexity of this allocation increases exponentially with the number of rounds, $T$. We conclude this section by discussing efficient methods to find an approximately optimal allocation. Since our focus is not on approximating the integrals here, we assume we have access to an oracle which can compute integrals and solve integral equations.

**Assumption 3.** *We have access to an oracle that, for any $i \in [n]$ and $t \in [T]$, can compute the integrals $\int_G \phi_i^t(v_i^t), dF_{max}^t(v_1, v_2)$ and $\int_G \frac{1 - F_i^t(v_i)}{f_i^t(v_i)}, dF_{max}^t(v_1, v_2)$ with $F_{max}^t(v_1, v_2) = F_1^t(v_1)^n F_2^t(v_2)^n$ and over any affine subspace $G$. Furthermore, the oracle can find the optimal allocation in the static case by solving the integral equations in Proposition 1.*

We start by making the following observation regarding the case $\delta = 1$.

**Fact 3.** *Suppose $\delta = 1$ and Assumptions 1-3 hold. Then, we can find the exact optimal allocation by calling the oracle $\mathcal{O}(T^2)$ times.*

To see why this holds, notice that, for the case of $\delta = 1$, the pair $(R_1^t, R_2^t)$ at most takes $T$ different values for any $t$, and so computing all $\{\mu^t(R_1^t, R_2^t), \nu_1^t(R_1^t, R_2^t), \nu_2^t(R_1^t, R_2^t)\}_t$ would overall require $\mathcal{O}(T^2)$ round of computation using the recursive derivations we stated earlier. However, for the case $\delta < 1$, this is not the case anymore, as the pair $(R_1^t, R_2^t)$ could take as much as $2^T$ different values. In this case, we can technically choose some $T_0 < T$ and aim to achieve the fairness constraint approximately in the first $T_0$ rounds, and then run the regular second-price auction in the remaining rounds. The following result formalizes this.

**Proposition 3.** *Suppose Assumptions 1-3 hold and that $\delta < 1$. Then, for any $\varepsilon \in [0, \min_i \alpha_i)$, there exists an allocation $\boldsymbol{x}'$ with the following properties: (1) $\boldsymbol{x}'$ guarantees that the fairness constraint for group $i$ is satisfied at a level of at least $(1 - \varepsilon)(\alpha_i - \varepsilon)$, (2) $\boldsymbol{x}'$ guarantees that the sellers total utility is at least that of the optimal allocation, and (3) $\boldsymbol{x}'$ can be computed by calling the oracle $\mathcal{O}\left(\varepsilon^{-1/\log(1/\delta)}\right)$ times.*

Intuitively, the approximation guarantees that the fairness constraint is satisfied within $\varepsilon$ of the target level by conducting the fair allocation scheme at level $\alpha_i - \varepsilon$ up to a sufficiently large time step such that approximate fairness is met no matter the allocations of the remaining rounds. In the remaining rounds, a standard second-price auction is carried out, thereby ensuring that total seller utility is at least that of the case where the fair allocation scheme is carried out to the last round.

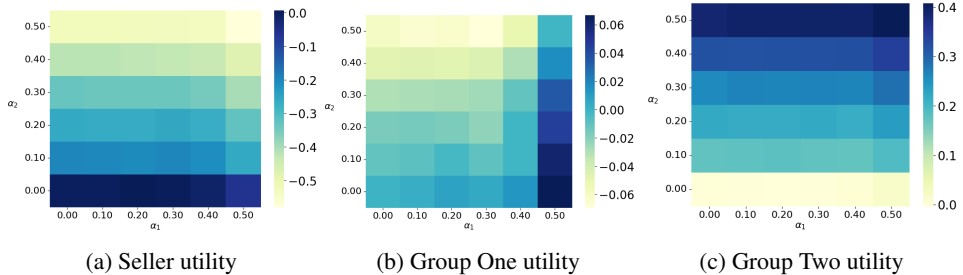

(a) Seller utility       (b) Group One utility       (c) Group Two utility

Figure 2: Difference in utility relative to unconstrained optimal allocation for $T = 2$

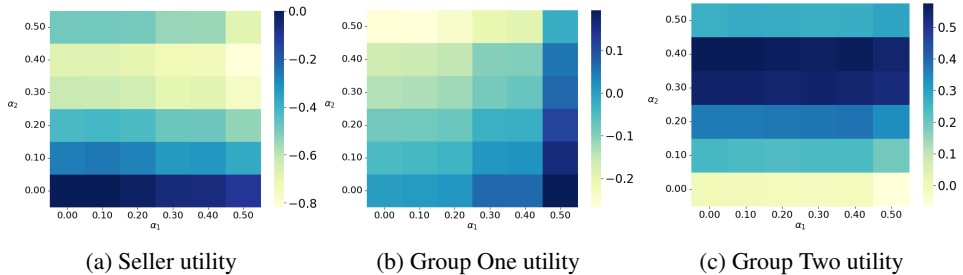

(a) Seller utility       (b) Group One utility       (c) Group Two utility

Figure 3: Difference in utility relative to unconstrained optimal allocation for $T = 4$

Notice that the complexity of the approximation given by proposition 3 grows large for $\delta$ close to 1. Next, we present a second approximation scheme for the case where $\delta$ is close to one.

**Proposition 4.** *Suppose Assumptions 1-3 hold and that $\delta < 1$. Then, for a fixed constant $c < \delta^2$, there exists an allocation $\boldsymbol{x}''$ with the following properties: $(1)$ $\boldsymbol{x}''$ satisfies the fairness constraint for group $i$ at a level of at least $c \cdot \alpha_i$, $(2)$ $\boldsymbol{x}''$ guarantees that the sellers total utility is at least $c$ times that of that of the optimal allocation, and $(3)$ $\boldsymbol{x}''$ can be computed by calling the oracle $Poly\left(\frac{1}{1-\delta}\right)$ times where the polynomial's degree and coefficients depend on $c$.*

This $\boldsymbol{x}''$ modifies the approximation $\boldsymbol{x}'$ presented in Proposition 3 by introducing a discontinuous discounting technique. In $\boldsymbol{x}''$, we partition the first $T_0$ rounds into buckets that use a common approximate discount factor such that the computational complexity of each bucket may be controlled in the same manner as Fact 3. A complete proof as well as a complete proof statement that relates the early stopping approximation with the discounting approximation that can be tuned to achieve an overall approximation level $c$ can be found in Appendix A.8.

## 5 Experiments

Here, we present the results of a numerical experiment assessing the impact on utilities of varying the fairness constraints of each group. In particular, we implement the case where $\delta = 0.99$, $n = 1$, $v_1^t \sim \text{Uniform}(0.5, 1.5)$, and $v_2^t \sim \text{Uniform}(0, 1)$ for $t \in [2]$ for $T = 2, 4$.

For each value of $T$, we consider combinations of fairness constraints on a discretized grid where $\alpha_1$ and $\alpha_2$ range over $(0, 0.5)$ with increments of $0.1$. For each pair $\alpha_1, \alpha_2$, we calculate the mean difference in utility between the optimal fair allocation at level $\alpha_1, \alpha_2$ and the unconstrained optimal allocation satisfying Assumption 2 (i.e., $\alpha_1, \alpha_2 = 0$), for the seller and buyers over 10,000 iterations of the mechanism. Note that, for these distributions, when $T = 2$ the average unconstrained allocation probabilities are $0.69$ and $0.31$ for groups one and two, respectively. The results are reported in Figures 2-3.

We see that the seller utility is decreasing in $\alpha_1, \alpha_2$ but only after the point at which the constraints bind. For group one, we see that, for a fixed $\alpha_1$, utility is decreasing in $\alpha_2$, since, to satisfy the fairness constraint, we re-allocate higher value regions to group two and allocate lower-value regions that would otherwise go unallocated to group one. In contrast, for group two, we see that, for a fixed $\alpha_2$, utility is relatively constant in $\alpha_1$ since the optimal allocation tends to reallocate to group one from the lower-value no-allocation region.

# 6 Acknowledgements

The authors thank Scott Kominers, Rakesh Vohra, and anonymous reviewers for insightful discussion and comments. Alireza Fallah acknowledges support from the European Research Council Synergy Program, the National Science Foundation under grant number DMS-1928930, and the Alfred P. Sloan Foundation under grant G-2021-16778. The latter two grants correspond to his residency at the Simons Laufer Mathematical Sciences Institute (formerly known as MSRI) in Berkeley, California, during the Fall 2023 semester. Michael Jordan acknowledges support from the Vannevar Bush Faculty Fellowship program under grant number N00014-21-1-2941 and the European Research Council (ERC-2022-SYG-OCEAN-101071601). Annie Ulichney's work is supported by the National Science Foundation Graduate Research Fellowship Program under Grant No. DGE 2146752. Any opinions, findings, and conclusions or recommendations expressed in this material are those of the author(s) and do not necessarily reflect the views of the National Science Foundation or the European Research Council.

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

# A  Appendix

In this appendix, we present proofs and details that are omitted from the body of the paper.

## A.1  Proof of Theorem 1

We first revisit the following well-known result for EPIC and IR mechanisms in the static scenario (see Chapters 9 and 13 of [30] for the proof):

**Proposition 5** ([28]). *A mechanism $(\boldsymbol{x}, \boldsymbol{p})$ is EPIC if and only if, for every $i$ and $k$, the allocation function $x_{i,k}(v, \boldsymbol{v}_{-(i,k)})$ is weakly increasing in $v$, and the payment function is given by*

$$p_{i,k}(v, \boldsymbol{v}_{-(i,k)}) = v\, x_{i,k}(v, \boldsymbol{v}_{-(i,k)}) - \int_{\underline{v}_i}^{v} x_{i,k}(z, \boldsymbol{v}_{-(i,k)}) dz + c(\boldsymbol{v}_{-(i,k)}), \qquad (13)$$

*with $c(\boldsymbol{v}_{-(i,k)}) = 0$ for IR mechanisms. Moreover, the seller's expected utility for any EPIC and IR mechanism is given by*

$$\mathbb{E}_{\boldsymbol{v}}\left[ \sum_{i\in[2],k\in[n]} \phi_i(v_{i,k}) x_{i,k}(\boldsymbol{v}) \right]. \qquad (14)$$

Next, let $\mathcal{S}$ be the set of $(\phi_i(v_{i,k}))_{i,k}$ for which the seller allocates the item to one of the groups in the optimal allocation $\tilde{x}$, i.e.,

$$\mathcal{S} = \left\{ (\phi_i(v_{i,k}))_{i,k} \,\Big|\, \exists i, k \text{ s.t. } x_{i,k}(\boldsymbol{v}) = 1 \right\}. \qquad (15)$$

Recall that $v_i = \max_k v_{i,k}$. We first establish that, given $\mathcal{S}$, the boundary of the optimal allocation is in the form of $\phi_1(v_1) = \phi_2(v_2) + \gamma$ for some $\gamma$. Note that the set $\mathcal{S}$ is measurable as the allocation function is measurable.

Notice that the probability of $\mathcal{S}$ is at least $\alpha_1 + \alpha_2$. Since $f_1(\cdot)$ and $f_2(\cdot)$ are continuous, we can find an allocation rule that is confined to $\mathcal{S}$, satisfies the fairness constraint, and its boundary takes the form $\phi_1(v_1) = \phi_2(v_2) + \gamma$ for some $\gamma$ (we call such allocations as *affine allocations*). That is, group one receives the object if and only if $\phi_1(v_1) \geq \phi_2(v_2) + \gamma$ and $(\phi_1(v_1), \phi_2(v_2)) \in \mathcal{S}$; similarly, group two receives the object if and only if $\phi_1(v_1) < \phi_2(v_2) + \gamma$ and $(\phi_1(v_1), \phi_2(v_2)) \in \mathcal{S}$. We further assume this allocation allocates the item to the buyer with highest value within each group. Denote this allocation by $x^*$. If there are multiple allocation rules of this form, we select the one whose corresponding $\gamma$ has the smallest absolute value. On can verify by inspection that such allocation is monotone, and hence, satisfies the EPIC condition with its corresponding payment identity. Without loss of generality, we may assume $\gamma > 0$, as the case for $\gamma < 0$ can be argued similarly.

If we had to allocate every pair in $\mathcal{S}$ but there were no fairness constraints, the optimal allocation would have been by checking to the buyer with the highest value, which from groups' pointw of view would mean the boundary $\phi_1(v_1) \lessgtr \phi_2(v_2)$. Let $A^*$ be the region that $x^*$ allocates differently from this unconstrained allocation over $\mathcal{S}$, i.e.,

$$A^* = \left\{ (\phi_i(v_{i,k}))_{i,k} \in \mathcal{S} \,\Big|\, \phi_1(v_1) \geq \phi_2(v_2) \geq \phi_1(v_1) - \gamma \right\}. \qquad (16)$$

We define $\tilde{A}$ similarly for the allocation $\tilde{x}$ as the set of values for which we allocate the item suboptimally. Also, for any $j \in [2]$, let $Q_j \subset \mathcal{S}$ be the region that is allocated to group $j$ in the optimal unconstrained allocation, i.e.,

$$Q_j = \left\{ (\phi_i(v_{i,k}))_{i,k} \in \mathcal{S} \,\Big|\, \phi_j(v_j) \geq \phi_{-j}(v_{-j}) \right\}. \qquad (17)$$

In particular, note that $A^* \subseteq Q_1$. We next make the following claim.

**Claim 1.** *The probability of $\tilde{A} \cap Q_1$ is lower bounded by the probability of $A^*$, i.e.,*

$$\mathbb{P}_{\boldsymbol{v}}(\tilde{A} \cap Q_1) \geq \mathbb{P}_{\boldsymbol{v}}(A^*). \qquad (18)$$

*Moreover, equality can only occur if $\tilde{A} \subseteq Q_1$.*

*Proof.* To see why this is the case, notice that since $\tilde{x}$ satisfies the fairness constraint, we should have

$$\mathbb{P}_{\boldsymbol{v}}(\tilde{A} \cap Q_1) + \mathbb{P}_{\boldsymbol{v}}(Q_2) \geq \mathbb{P}_{\boldsymbol{v}}(\tilde{A} \cap Q_1) + \mathbb{P}_{\boldsymbol{v}}(Q_2 \backslash \tilde{A}) \geq \alpha_2. \tag{19}$$

Now, if (18) does not hold, then we would have

$$\mathbb{P}_{\boldsymbol{v}}(A^*) + \mathbb{P}_{\boldsymbol{v}}(Q_2) > \alpha_2, \tag{20}$$

but then this would mean that we can lower the $\gamma$ in allocation $x^*$ which contradicts its definition. $\quad\square$

Notice that the loss of allocation $x^*$ compared to the optimal unconstrained allocation is given by

$$\text{Loss}_{x^*} := \int_{A^*} (\phi_1(\max_k v_{1,k}) - \phi_2(\max_k v_{2,k})) dF(\boldsymbol{v}) \tag{21}$$

Let also $\text{Loss}_{\tilde{x}}$ denote the loss of allocation $\tilde{x}$ compared to the optimal unconstrained allocation. This loss is lower bounded by

$$\text{Loss}_{\tilde{x}} \geq \int_{\tilde{A} \cap Q_1} (\phi_1(\max_k v_{1,k}) - \phi_2(\max_k v_{2,k})) dF(\boldsymbol{v}). \tag{22}$$

Notice that, since $\tilde{x}$ is the optimal allocation, the right hand side of (22) should be lower than (21):

$$\int_{\tilde{A} \cap Q_1} (\phi_1(\max_k v_{1,k}) - \phi_2(\max_k v_{2,k})) dF(\boldsymbol{v}) \leq \int_{A^*} (\phi_1(\max_k v_{1,k}) - \phi_2(\max_k v_{2,k})) dF(\boldsymbol{v}). \tag{23}$$

We next claim that this implies that $A^* = \tilde{A} \cap Q_1$.

**Claim 2.** *We have $A^* = \tilde{A} \cap Q_1$ (up to a measure-zero set).*

*Proof.* To simplify the notation, let $L(v_1, v_2) = \phi_1(v_1) - \phi_2(v_2)$. Suppose these two sets are not equal. This implies that there is some region $\tilde{B}$ outside region $A^*$ that is in region $\tilde{A} \cap Q_1$ and some region $B^*$ that is outside region $\tilde{A} \cap Q_1$ but inside region $A^*$. By Claim 1, we have:

$$\int_{\tilde{B}} dF(\boldsymbol{v}) \geq \int_{B^*} dF(\boldsymbol{v}). \tag{24}$$

Next, by (23), we have

$$\int_{\tilde{B}} L(v_1, v_2) dF(\boldsymbol{v}) \leq \int_{B^*} L(v_1, v_2) dF(\boldsymbol{v}) \tag{25}$$

Notice that $L(v_1, v_2)$ is nonnegative over $\tilde{B}$ and $B^*$. Also, the fact that $\tilde{B} \cap A^*$ is empty (along with $\tilde{B} \subseteq Q_1$) means that $\sup_{B^*} L(v_1, v_2) \leq \gamma < \inf_{\tilde{B}} L(v_1, v_2)$. Next, notice that

$$\int_{B^*} L(v_1, v_2) dF(\boldsymbol{v}) \leq \sup_{B^*} L(v_1, v_2) \int_{B^*} dF(\boldsymbol{v})$$

and

$$\int_{\tilde{B}} L(v_1, v_2) dF(\boldsymbol{v}) \geq \inf_{\tilde{B}} L(v_1, v_2) \int_{\tilde{B}} dF(\boldsymbol{v}).$$

Combined with (24), these inequalities yield that

$$\int_{B^*} L(v_1, v_2) dF(\boldsymbol{v}) \leq \int_{\tilde{B}} L(v_1, v_2) dF(\boldsymbol{v}),$$

where the equality holds only if $\tilde{B}$ and $B^*$ are measure zero, which should be the case given (25). This shows that $\tilde{x}$ and $x^*$ are the same, up to a measure zero set. $\quad\square$

This claim along with the equality condition of Claim 1 shows that $A^*$ and $\tilde{A}$ are equal up to a measure-zero set. Hence, for a given $\mathcal{S}$, the optimal allocation is an affine allocation.

Now, this result was for a given $\mathcal{S}$. Using a similar argument, we can establish that the optimal set $\mathcal{S}$ should be in the form of

$$\left\{ (\phi_1(v_1), \phi_2(v_2)) \Big| \phi_1(v_1) \geq -\eta_1 \text{ or } \phi_2(v_2) \geq -\eta_2 \right\} \tag{26}$$

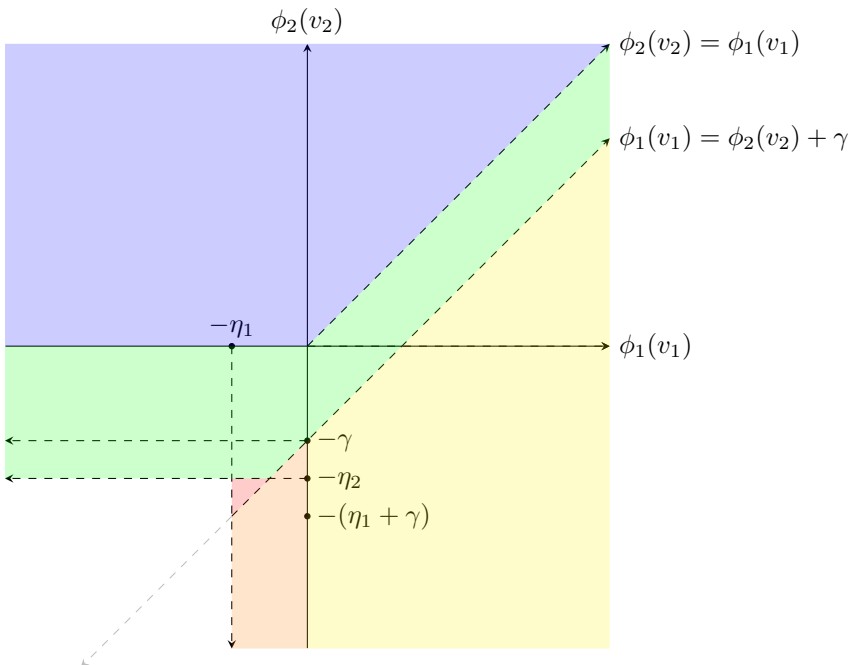

Figure 4: The case where $\eta_2 < \eta_1 + \gamma$.

for some $\eta_1, \eta_2 \geq 0$. We finally establish that $\eta_2 = \eta_1 + \gamma$. Suppose that $\eta_2 < \eta_1 + \gamma$ in the optimal fair allocation. Then, there exists the following region of allocations to group one: $G = \{(\phi_1(v_1), \phi_2(v_2)) : \phi_2(v_2) \in (-\eta_2, -\eta_1 - \gamma), \phi_1(v_1) \in (-\eta_1, -\eta_1 - \gamma + \eta_2), \phi_2(v_2) \geq \phi_1(v_1) + \gamma\}$. Notice that $G$ constitutes a triangle as depicted in the red region in Figure 4. Consider modifying the allocation rule as follows. For some $g \in G$, do not allocate $g$. Instead, allocate a region of equal measure to group 1 beginning at $(-\eta_1, -\eta_2)$. This switch changes the loss by the amount $\eta_1 - \eta_2$. Then, we return to the original and total levels of allocation by reallocating another equal-measure region on the border $\phi_1(v_1) = \phi_2(v_2) - \gamma$ to group 2. The second switch changes the loss by the amount $\gamma$. Therefore, the total change to the loss is $\eta_1 - \eta_2 + \gamma < 0$ by our initial assumption. Thus, this modification results in decreased loss to the seller, and it cannot be optimal. Now, suppose that $\eta_2 > \eta_1 + \gamma$ in the optimal fair allocation. Then, there exists a region that is not allocated under the optimal mechanism defined by $H = \{(\phi_1(v_1), \phi_2(v_2)) : \phi_2(v_2) \in (-\eta_1 - \gamma, -\eta_1), \phi_1(v_1) \in (-\eta_2 - \gamma, -\eta_1), \phi_2(v_2) \leq \phi_1(v_1) + \gamma\}$. As before, $H$ is a triangle depicted in gray in Figure 5. However, notice that the unallocated values in $H$ are closer to the unconstrained allocation boundary than the allocated set of virtual values $\{(\phi_1(v_1), \phi_2(v_2)) : \phi_1(v_1) <= \eta_1, \phi_2(v_2) \in (-\eta_1 - \gamma, -\eta_2), \phi_2(v_2) \geq \phi_1(v_1) - \gamma\}$. Therefore, such an allocation cannot be the optimal fair allocation due to the result shown in the proof of Theorem 1 that loss is increasing in Euclidean distance from the unconstrained allocation boundary (1a). Observing that these two cases together imply equality concludes the proof. ∎

### A.2  Proof of proposition 1

Let $\text{Loss}_x$ and $\text{Loss}_{\widehat{x}}$ denote the loss of allocations $x$ and $\tilde{x}$ compared to the original unconstrained allocation, respectively. Notice that the change in loss under the modified allocation can be expressed as

$$\text{Loss}_x - \text{Loss}_{\widehat{x}} = \int_{\widehat{G}_2} (\phi_1(\max_k v_{1,k}) - \phi_2(\max_k v_{2,k})) dF(\boldsymbol{v}). \tag{27}$$

Since $\phi_1(\max_k v_{1,k}) - \phi_2(\max_k v_{2,k}) \in (0, \gamma)$ for every pair $(\max_k v_{1,k}, \max_k v_{2,k}) \in \widehat{G}_2$ and $\widehat{G}_2$ is a set with nonzero measure, we may conclude that $\text{Loss}_x - \text{Loss}_{\widehat{x}} > 0$ In other words, the seller's utility increases under the allocation $\widehat{x}$ as compared to that of allocation $x$. Therefore, for an optimal fair allocation where $\gamma > 0$, it must be that $\mathbb{P}(G_2) = \alpha_2$.

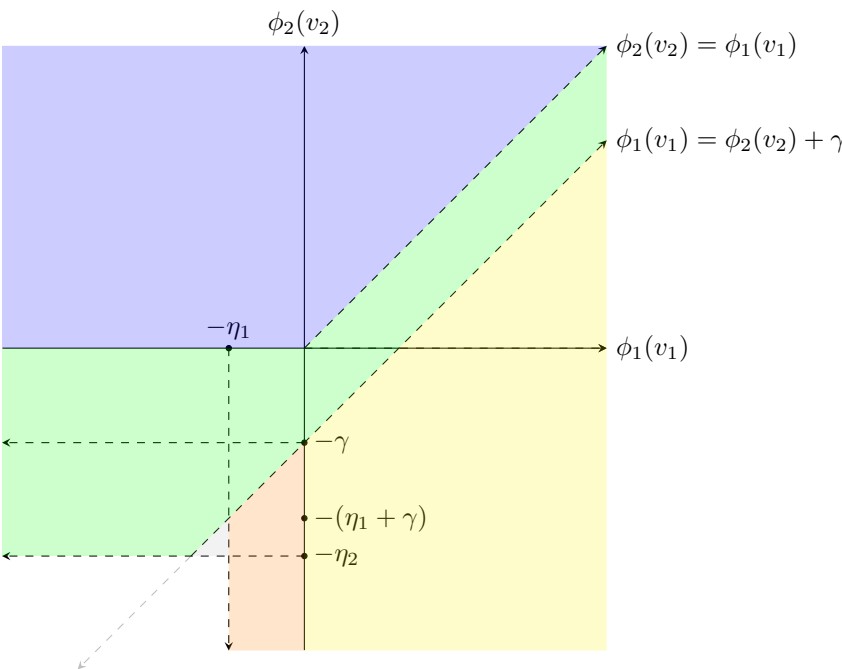

Figure 5: The case where $\eta_2 > \eta_1 + \gamma$.

The proof is similar to show that, if $\eta_1 > 0$, i.e., there is insufficient total allocation in the optimal unconstrained allocation, we have $\mathbb{P}(G_1) = \alpha_1$. As before, clearly $\mathbb{P}(G_1) \geq \alpha_1$, otherwise the fairness constraint is not satisfied. Suppose, for sake of contradiction, that $\mathbb{P}(G_1) \geq \alpha_1$. Then there is some subset

$$\widehat{G}_1 \subseteq \{(v_1, v_2) | -\eta_1 \leq \phi_1(v_1), \leq 0, \phi_2(v_2) \leq \phi_1(v_1) - \gamma\} \tag{28}$$

with nonzero measure such that $\mathbb{P}(G_1 \setminus \widehat{G}_2) \geq \alpha_1$. Notice that $\widehat{G}_1$ is some subset of the orange region in Figure 1b. Once again, let $x$ be the original allocation and $\widehat{x}$ be the modified allocation where we do not allocate region $\widehat{G}_2$. The change in loss under the modified allocation is given by

$$\text{Loss}_x - \text{Loss}_{\widehat{x}} = \int_{\widehat{G}_1} (\phi_1(\max_k v_{1,k}) - \phi_2(\max_k v_{2,k})) dF(\boldsymbol{v}). \tag{29}$$

Since $\widehat{G}_1$ has nonzero measure and $\phi_2(v_2) \leq 0$ for all $(\max_k v_{1,k}, \max_k v_{2,k}) \in \widehat{G}_1$, it follows that $\text{Loss}_{\widehat{x}} - \text{Loss}_x > 0$. Once again, this contradicts the optimality of allocation $x$, and it must be that $\mathbb{P}(G_1) = \alpha_1$.

Observing that the second claim follows from exchanging the roles of groups 1 and 2 in the preceding work concludes the proof. ∎

### A.3 Proof of corollary 1

Both results follow from an application of the following claim.

**Claim 3.** *Let $f(v_{i,k}^T)$ be some function of $v_{i,k}^T$. Then*

$$\mathbb{E}\left[x_{i,k}^T(\boldsymbol{b}^T, \boldsymbol{h}^T)f(v_{i,k}^T)\right] = \frac{1}{n}\int_{G_i} f(v_{i,k}^T)dF_{max}^T(v_1, v_2) \tag{30}$$

First, by the law of total probability and observing that $x_{i,k}^T = 0$ if $v_{i,k}^T < \max_k v_{i,k}^T$,

$$\mathbb{E}\left[x_{i,k}^T(\boldsymbol{b}^T, \boldsymbol{h}^T)f(v_{i,k}^T)\right] = \mathbb{E}\left[x_{i,k}^T(\boldsymbol{b}^T, \boldsymbol{h}^T)f(v_{i,k}^T) \mid v_{i,k}^T = \max_k v_{i,k}^T\right]\mathbb{P}\left(v_{i,k}^T = \max_k v_{i,k}^T\right) \tag{31}$$

Next, since $v_{i,k}^T \overset{i.i.d.}{\sim} F_i^T$, $\mathbb{P}(v_{i,k}^T = \max_k v_{i,k}) = \frac{1}{n}$. Also observe that $x_{i,k}^T = 0$ for $(v_1, v_2) \notin G_i$. Thus,

$$
\begin{aligned}
\mathbb{E}\left[x_{i,k}^T(\boldsymbol{b}^T, \boldsymbol{h}^T)f(v_{i,k}^T)\right] &= \frac{1}{n}\mathbb{E}\left[f(v_{i,k}^T)\mathbb{1}\{v_{i,k}^T = \max_k v_{i,k}^T, (v_1, v_2) \in G_i\}\right] \\
&= \frac{1}{n}\int_{G_i} f(v_{i,k}^T)dF_{\max}^T(v_1, v_2)
\end{aligned}
\tag{32}
$$

Now, we use the preceding claim to evaluate the expected utility of a buyer in group $i$ denoted $\nu_i^T(R_1^T, R_2^T)$. Note that

$$
\nu_i^T(R_1^T, R_2^T) = \mathbb{E}\left[\mathcal{U}_{i,k}^T(v_{i,k}^T; \boldsymbol{b}^T, \boldsymbol{h}^T)\right] = \mathbb{E}\left[x_{i,k}^T(\boldsymbol{b}^T, \boldsymbol{h}^T)v_{i,k}^T - p_{i,k}^T(\boldsymbol{b}^T, \boldsymbol{h}^T)\right]
\tag{33}
$$

The expected payment of a buyer in group $i$ can equivalently be expressed as

$$
\mathbb{E}\left[p_{i,k}^T(\boldsymbol{b}^T, \boldsymbol{h}^T)\right] = \mathbb{E}\left[x_{i,k}^T(\boldsymbol{b}^T, \boldsymbol{h}^T)\phi_i^T(v_{i,k}^T)\right]
\tag{34}
$$

as shown in [30, Lemma 13.11]. Substitution of this result into  yields

$$
\nu_i^T(R_1^T, R_2^T) = \mathbb{E}\left[x_{i,k}^T(\boldsymbol{b}^T, \boldsymbol{h}^T)\left(v_{i,k}^T - \phi_i^T(v_{i,k}^T)\right)\right]
\tag{35}
$$

From here, applying the claim 3 to the function $v_{i,k}^T - p_{i,k}^T(\boldsymbol{b}^T, \boldsymbol{h}^T)$ yields the desired result.

Next, we show the second part of the claim. By definition 14, we may express the expected utility of the seller in group $i$ denoted $\mu^T(R_1^T, R_2^T)$ as

$$
\mathbb{E}\left[\phi_1(v_{1,k})x_{1,k}(\boldsymbol{v}) + \phi_2(v_{2,k})x_{2,k}(\boldsymbol{v})\right]
\tag{36}
$$

By linearity of expectation and an application of claim 3 to functions $\phi_1(v_{1,k})$ and $\phi_2(v_{2,k})$ yields the desired result. ∎

## A.4 Proof of proposition 2

First, notice that the seller must allocate the item to a member of the group, otherwise the auction becomes infeasible. Therefore, this case reduces to a static Vickrey auction with no reserve price where the revenue-maximizing allocation and payments are characterized by [28] as follows. To optimize seller revenue, the seller allocates to the highest bidder who pays an amount equal to the second highest bid.

By definition, we may express the expected utility as

$$
\nu_i^t(R_1^t, R_2^t) = \mathbb{E}\left[\sum_{\tau=t}^T \delta^{\tau-t}\mathcal{U}_{i,k}^\tau\right] = \mathbb{E}\left[\mathcal{U}_{i,k}^t\right] + \delta\mathbb{E}\left[\sum_{\tau=t+1}^T \delta^{\tau-(t+1)}\mathcal{U}_{i,k}^\tau\right]
\tag{37}
$$

Since the seller allocates the item to a buyer in group $i$ with probability 1 to maintain feasibility and by Fact 1,

$$
\delta\mathbb{E}\left[\sum_{\tau=t+1}^T \delta^{\tau-(t+1)}\mathcal{U}_{i,k}^\tau\right] = \delta\nu_i^{t+1}((R_i^t - 1)/\delta, R_{-i}^t/\delta)
\tag{38}
$$

From (35), the definition of virtual values, and the observation that this setting reduces to a Vickrey auction,

$$
\mathbb{E}\left[x_{i,k}^t(\boldsymbol{b}^t, \boldsymbol{h}^t)\left(v_{i,k}^t - \phi_i^t(v_{i,k}^t)\right)\right] = \mathbb{E}\left[\frac{1 - F_i^t(v_{i,k}^t)}{f_i^t(v_{i,k}^t)}\mathbb{1}\left\{v_{i,k}^t = \max_j v_{i,j}^t\right\}\right]
\tag{39}
$$

Since $v_{i,k}^t$ is distributed according to $F_i^t(v_{i,k}^t)$ for all $k \in [n]$, $\mathbb{P}\left(v_{i,k}^t = \max_j v_{i,j}^t\right) = (F_i^t(v))^{n-1}$. Therefore,

$$
\mathbb{E}\left[x_{i,k}^t(\boldsymbol{b}^t, \boldsymbol{h}^t)\left(v_{i,k}^t - \phi_i^t(v_{i,k}^t)\right)\right] = \int(1 - F_i^t(v))(F_i^t(v))^{n-1}dv
\tag{40}
$$

and the result follows.

Next, we evaluate the expected utility of buyer $-i$ that does not receive the item. Since buyer $-i$ does not receive the item with probability 1, their expected utility reduces to the discounted expected utility over future rounds, i.e.,

$$\nu_{-i}^t(R_1^t, R_2^t) = \delta\mathbb{E}\left[\sum_{\tau=t+1}^{T}\delta^{\tau-(t+1)}\mathcal{U}_{i,k}^\tau\right]. \tag{41}$$

The result follows by applying Fact 1 to see that

$$\delta\mathbb{E}\left[\sum_{\tau=t+1}^{T}\delta^{\tau-(t+1)}\mathcal{U}_{i,k}^\tau\right] = \delta\nu_{-i}^{t+1}((R_1^t-1)/\delta, R_2^t/\delta). \tag{42}$$

Last, we evaluate the seller's expected utility. By Definition 14 and since the good is allocated to the maximum valuation buyer in group $i$ with probability 1,

$$\mu^t(R_1^t, R_2^t) = \mathbb{E}\left[\sum_{\tau=t}^{T}\delta^{\tau-t}\left[\sum_{i\in[2],k\in[n]}\phi_i^\tau(v_{i,k}^\tau)x_{i,k}^\tau(\boldsymbol{v})\right]\right] = \mathbb{E}\left[\phi_i^t\left(\max_k v_{i,k}^t\right)\right] + \delta\mu^{t+1}((R_i^t-1)/\delta, R_{-i}^t/\delta). \tag{43}$$

Recalling that $\max_k v_{i,k}^t \sim F_i^t(v)^n$, we can express the first term as

$$\mathbb{E}\left[\phi_i^t\left(\max_k v_{i,k}^t\right)\right] = \int \phi_i^t(v)dF_i^t(v)^n \tag{44}$$

and the result follows. ∎

### A.4.1 Update of interim functions

Under the premise of Proposition 2, the interim functions are updates in the following manner:

$$\nu_i^t(R_1^t, R_2^t) = \int(1-F_i^t(v))(F_i^t(v))^{n-1}dv + \delta\nu_i^{t+1}((R_i^t-1)/\delta, R_{-i}^t/\delta), \tag{45a}$$

$$\nu_{-i}^t(R_1^t, R_2^t) = \delta\nu_{-i}^{t+1}((R_i^t-1)/\delta, R_{-i}^t/\delta), \tag{45b}$$

$$\mu^t(R_1^t, R_2^t) = \int \phi_i^t(v)dF_i^t(v)^n + \delta\mu^{t+1}((R_i^t-1)/\delta, R_{-i}^t/\delta). \tag{45c}$$

### A.5 Example with negative net utility of not receiving an item

One might expect the net utility of not receiving the item to be nonnegative for each group, given that not receiving the item at the current time implies a higher chance of receiving it in future rounds. However, interestingly, this argument is not necessarily true; the expected utility of a buyer may decrease even as the fairness constraint on their group increases. Let us elaborate this matter by a simple example. Consider a scenario with $T = 2$, $n = 1$, and $\delta = 0.8$. Suppose the distributions of values are identical across both groups, i.e., $F_1^t = F_2^t$ for $t \in [2]$, while the values from the first round are significantly lower than those from the second round, i.e., $\bar{v}_i^1 \ll \underline{v}_i^2$. Initially, let $\alpha_1 = \alpha_2 = 0.2$. In this scenario, each group has some positive probability of receiving the item in the second round. Now, let us increase $\alpha_1$ to $0.5$. It is straightforward to verify that the only feasible allocation is to give the item to the first group in the first round and to the second group in the second round. Given that the values in the first round are considerably lower, this allocation actually decreases the expected utility for the first group.

### A.6 Proof of theorem 2

Notice that the variable

$$\sum_{\ell=1}^{n}x_{i,\ell}^t(\boldsymbol{v}^t). \tag{46}$$

determines whether group $i$ receives the item or not. Now, suppose buyer $(i, k)$ submits bid $b$. Given Assumption 2, we can write the utility of buyer $(i, k)$ as:

$$x_{i,k}^t(b, \boldsymbol{v}_{-(i,k)}^t)v_{i,k}^t - p_{i,k}^t(b, \boldsymbol{v}_{-(i,k)}^t) - \delta(\sum_{\ell=1}^n x_{i,\ell}^t(b, \boldsymbol{v}_{-(i,k)}^t))\Delta_i^t(R_1^t, R_2^t) + \delta\nu_i^{t+1}(R_i^t/\delta, (R_{-i}^t-1)/\delta).$$
(47)

Notice that the last term $\delta\nu_i^{t+1}(R_i^t/\delta, (R_{-i}^t - 1)/\delta)$ is a constant, independent of the values and the bid. Hence, we can drop this constant from the buyer's decision making. Now, assuming $\boldsymbol{v}_{-(i,k)}^t$ is fixed, we can write the adjusted utility as

$$x_{i,k}^t(b, \boldsymbol{v}_{-(i,k)}^t)v_{i,k}^t - \tilde{p}_{i,k}^t(b, \boldsymbol{v}_{-(i,k)}^t),$$
(48)

where

$$\tilde{p}_{i,k}^t(b, \boldsymbol{v}_{-(i,k)}^t) := p_{i,k}^t(b, \boldsymbol{v}_{-(i,k)}^t) + \delta(\sum_{\ell=1}^n x_{i,\ell}^t(b, \boldsymbol{v}_{-(i,k)}^t))\Delta_i^t(R_1^t, R_2^t).$$
(49)

Using this representation, we can interpret round $t$-th auction as a one round auction. Hence, given proposition 5, for an allocation to be EPIC, $x_{i,k}^t(b, \boldsymbol{v}_{-(i,k)}^t)$ should be (weakly) monotone in $b$ and we also should have

$$\tilde{p}_{i,k}^t(b, \boldsymbol{v}_{-(i,k)}^t) = b\, x_{i,k}^t(b, \boldsymbol{v}_{-(i,k)}^t) - \int_{\underline{v}_i^t}^b x_{i,k}^t(z, \boldsymbol{v}_{-(i,k)}^t)dz + c_{i,k}(\boldsymbol{v}_{-(i,k)}^t),$$
(50)

for some function $c_{i,k}(\cdot)$ which is independent of $v$. As a result, for any $i \in [2]$ and $k \in [n]$, the payment is given by

$$p_{i,k}^t(v, \boldsymbol{v}_{-(i,k)}^t) =$$
(51)
$$v\, x_{i,k}^t(v, \boldsymbol{v}_{-(i,k)}^t) - \int_{\underline{v}_i^t}^v x_{i,k}^t(z, \boldsymbol{v}_{-(i,k)}^t)dz - \delta(\sum_{\ell=1}^n x_{i,\ell}^t(v, \boldsymbol{v}_{-(i,k)}^t))\Delta_i^t(R_1^t, R_2^t) + c_{i,k}(\boldsymbol{v}_{-(i,k)}^t).$$

The seller wants to choose $c_{i,k}(\cdot)$ as large as possible, but the IR constraint puts an upper bound on it. Substituting the payment (51) into (47), the buyer utility is given by

$$\int_{\underline{v}_i^t}^{v_{i,k}^t} x_{i,k}^t(z, \boldsymbol{v}_{-(i,k)}^t)dz + \delta\nu_i^{t+1}(R_i^t/\delta, (R_{-i}^t - 1)/\delta) - c_{i,k}(\boldsymbol{v}_{-(i,k)}^t).$$
(52)

Now, to check the IR constraint, suppose buyer $(i, k)$ skips round $t$, and denote the optimal allocation in that case by $\tilde{\boldsymbol{x}}^t$. Hence, the expected utility of buyer $(i, k)$ from time $t + 1$ onwards is given by

$$\delta\nu_i^{t+1}(R_i^t/\delta, (R_{-i}^t - 1)/\delta) - \delta(\sum_{\ell \neq k} \tilde{x}_{i,\ell}^t(\boldsymbol{v}_{-(i,k)}^t))\Delta_i^t(R_1^t, R_2^t).$$
(53)

The IR constraint implies that the expectation of (52) minus (53) over $\boldsymbol{v}_{-(i,k)}^t$ should be nonnegative for any $v_{i,k}^t$. Thus, we should have

$$\mathbb{E}_{\boldsymbol{v}_{-(i,k)}^t}[c_{i,k}(\boldsymbol{v}_{-(i,k)}^t)] \leq \delta\Delta_i^t(R_1^t, R_2^t)\, \mathbb{E}_{\boldsymbol{v}_{-(i,k)}^t}\left[\sum_{\ell \neq k} \tilde{x}_{i,\ell}^t(\boldsymbol{v}_{-(i,k)}^t)\right],$$
(54)

and so, the seller sets the function $c_{i,k}(\cdot)$ to achieve the equality case. Note that the right hand side is only a function of $i$. We denote it by $c_i^t$. We next make the following claim regarding the expected payment of buyer $(i, k)$ to the seller in this case.

**Claim 4.** *The expected payment of buyer $(i, k)$ to the seller at round $t$ is given by*

$$\mathbb{E}_{\boldsymbol{v}^t}\left[\phi_i^t(v_{i,k}^t)x_{i,k}^t(\boldsymbol{v}^t) - \delta(\sum_{\ell=1}^n x_{i,\ell}^t(\boldsymbol{v}^t))\Delta_i^t(R_1^t, R_2^t)\right] + c_i^t.$$
(55)

The proof of this claim follows from the classical derivation of payment in static mechanism design (see [30][Lemma 13.11] for instance) along with the characterization of payment in (51). As a consequence, the seller's utility at round $t$ is given by

$$\mathbb{E}_{\boldsymbol{v}^t}\left[\sum_{i=1}^{2}\sum_{k=1}^{n}\phi_i^t(v_{i,k}^t)x_{i,k}^t(\boldsymbol{v}^t) - \sum_{i=1}^{2}n\delta\Delta_i^t(R_1^t,R_2^t)\sum_{\ell=1}^{n}x_{i,\ell}^t(\boldsymbol{v}^t)\right] + nc_1^t + nc_2^t. \quad (56)$$

Therefore, the seller's expected utility for time $t$ onwards is given by

$$\mathbb{E}_{\boldsymbol{v}^t}\left[\sum_{i=1}^{2}\sum_{k=1}^{n}\phi_i^t(v_{i,k}^t)x_{i,k}^t(\boldsymbol{v}^t) + \sum_{i=1}^{2}\delta\left(\mu^{t+1}((R_i^t-1)/\delta, R_{-i}^t/\delta) - n\Delta_i^t(R_1^t,R_2^t)\right)\sum_{\ell=1}^{n}x_{i,\ell}^t(\boldsymbol{v}^t)\right] + nc_1^t + nc_2^t. \quad (57)$$

Notice that the seller wants to maximize (57). First, notice that, since $\phi_i^t(\cdot)$ is increasing by Assumption 1, if we allocate the item to group $i$, we would allocate it to the buyer with the highest value which we denote its value by $v_i^t = \max_k v_{i,k}^t$.

Now, we need to determine whether the item is allocated to the buyer with the highest value in group one or to the buyer with the highest value in the second group. Given (57), it is straightforward to see that the item is allocated to group one if

$$\phi_1^t(v_1^t) - \phi_2^t(v_2^t) \geq \quad (58)$$
$$n\delta\left(\Delta_1^t(R_1^t,R_2^t) - \Delta_2^t(R_1^t,R_2^t)\right) + \delta\left(\mu^{t+1}((R_2^t-1)/\delta, R_1^t/\delta) - \mu^{t+1}((R_1^t-1)/\delta, R_2^t/\delta)\right),$$

and otherwise it is allocated to the second group. Notice that this allocation is indeed monotone, and hence, along with the above payment, it is an EPIC and IR allocation. This gives us the set $G_i^t$ in the theorem's statement. Finally, to derive the update of interim functions, notice that the expected utility of seller follows from (57). The expected utility of buyers can also be derived from Claim 4 similar to the proof of Corollary 1.

### A.6.1 Characterization of $\zeta_i^t$

Recall that $c_i^t$ is given by

$$c_i^t = \delta\Delta_i^t(R_1^t,R_2^t)\zeta_i^t \quad (59)$$

where

$$\zeta_i^t = \mathbb{E}_{\boldsymbol{v}_{-(i,k)}^t}\left[\sum_{\ell\neq k}\tilde{x}_{i,\ell}^t(\boldsymbol{v}_{-(i,k)}^t)\right] \quad (60)$$

is the probability of group $i$ winning the object when they participate with one fewer buyer. Notice that, an argument similar to the one we made above implies that in the case where we have $n-1$ buyers from group $i$ and $n$ buyers from the other group, group $i$ wins the item if $(\max_{\ell\neq k} v_{i,\ell}^t, \max_\ell v_{(-i),\ell}^t) \in \tilde{G}_i^t$ with

$$\tilde{G}_i^t := \left\{(v_1,v_2)\,\Big|\,\phi_i^t(v_i) - \phi_{-i}^t(v_{-i}) \geq (n-1)\delta\Delta_i^t(R_1^t,R_2^t) - n\delta\Delta_{-i}^t(R_1^t,R_2^t) + (-1)^i\delta\Delta_0^t(R_1^t,R_2^t)\right\}.$$

As a result, we have

$$\zeta_i^t = \int_{\tilde{G}_i^t} d(F_i^t(v_i))^{n-1}(F_{-i}^t(v_{-i}))^n. \quad (61)$$

### A.6.2 Update of interim functions

The expected utility of seller follows from (57). The expected utility of buyers can also be derived from Claim 4 similar to the proof of Corollary 1. Hence, we have

$$\nu_i^t(R_1^t,R_2^t) = \frac{1}{n}\int_{G_i^t}\frac{1-F_i^t(v_i)}{f_i^t(v_i)}dF_{\max}^t(v_1,v_2) + \delta\nu_i^{t+1}(R_i^t/\delta, (R_{-i}^t-1)/\delta) - \delta\Delta_i^t(R_1^t,R_2^t)\zeta_i^t, \quad (62a)$$

$$\mu^t(R_1^t,R_2^t) = \quad (62b)$$

$$\sum_{i=1}^{2}\left(\int_{G_i^t}\phi_i^t(v_i^t)\,dF_{\max}^t(v_1,v_2) + \delta\mathbb{P}(G_i^t)\mu^{t+1}((R_i^t-1)/\delta, R_{-i}^t/\delta) + \delta n\Delta_i^t(R_1^t,R_2^t)(\zeta_i^t - \mathbb{P}(G_i^t))\right),$$

where the probability measure $\mathbb{P}(\cdot)$ is taken with respect to the distribution $F_{\max}^t(v_1,v_2) := F_1^t(v_1)^n F_2^t(v_2)^n$.

### A.6.3 Relaxing Assumption 2

Revisiting our proof shows that we can relax Assumption 2 and repeat the proof steps. In particular, when not allocating to either of the two groups is feasible, the utility of buyer $(i, k)$ in (47) will change to

$$x_{i,k}^t(b, \boldsymbol{v}_{-(i,k)}^t)v_{i,k}^t - p_{i,k}^t(b, \boldsymbol{v}_{-(i,k)}^t) + \delta\nu_i^{t+1}(R_i^t/\delta, R_{-i}^t/\delta)$$
$$- \delta(\sum_{\ell=1}^n x_{i,\ell}^t(b, \boldsymbol{v}_{-(i,k)}^t))\Delta_{i,i}'(R_1^t, R_2^t) - \delta(\sum_{\ell=1}^n x_{(-i),\ell}^t(b, \boldsymbol{v}_{-(i,k)}^t))\Delta_{i,-i}'(R_1^t, R_2^t), \quad (63)$$

with

$$\Delta_{i,i}'(R_1^t, R_2^t) = \nu_i^{t+1}(R_i^t/\delta, R_{-i}^t/\delta) - \nu_i^{t+1}((R_i^t - 1)/\delta, R_{-i}^t/\delta), \quad (64)$$
$$\Delta_{i,-i}'(R_1^t, R_2^t) = \nu_i^{t+1}(R_i^t/\delta, R_{-i}^t/\delta) - \nu_i^{t+1}(R_i^t/\delta, (R_{-i}^t - 1)/\delta) \quad (65)$$

We can follow similar steps to derive the payment and allocations. We do not repeat the proof steps here as they follow a similar argument.

### A.7 Proof of proposition 3

We first define $\boldsymbol{x}'$ as follows: We assume that the auction is designed to run for $T_0$ rounds (for a chosen value of $T_0$ determined later) and use the recursive functions developed earlier to determine the exact fair allocation under the fairness constraint at level $\alpha_i - \varepsilon$ for group $i$ (computed over the first $T_0$ rounds). For the remaining $T - T_0$ rounds, we conduct the standard second-price auction.

First, we characterize the approximation $\boldsymbol{x}'$ in relation to the exact fair allocation. Let $x_i^{*t}$ denote the allocation to group $i$ at time $t$ under the optimal allocation that satisfies the fairness constraint (2) at a level $\alpha_i$ for $i \in \{1, 2\}$. Next, we consider the fairness guarantees of $\boldsymbol{x}^* := \{x_i^{*t}\}_{t=1}^T$ with early stopping.

**Claim 5.** *If the allocation $\boldsymbol{x}^*$ is stopped at time $T_0 := \frac{\delta}{\varepsilon} < T$, then $\{x_i^{*t}\}_{t=1}^{T_0}$ satisfies the fairness constraint (2) at a level $(\alpha_i - \varepsilon)$.*

*Proof.* Observe that $\boldsymbol{x}^*$ satisfies

$$\sum_{t=1}^T \delta^{t-1} x_i^{*t} = \sum_{t=1}^{T_0} \delta^{t-1} x_i^{*t} + \sum_{t=T_0+1}^T \delta^{t-1} x_i^{*t} \geq \alpha_i \sum_{t=0}^{T-1} \delta^t. \quad (66)$$

Now, since $x_i^{*t} \in \{0, 1\}$ for all $t$, we can see that

$$\sum_{t=T_0}^T \delta^{t-1} x_i^{*t} \leq \delta^{T_0} \sum_{t=0}^{T-1-T_0} \delta^t = \varepsilon \sum_{t=0}^{T-1-T_0} \delta^t$$

where the second equality follows from the fact that $\delta^{T_0} = \varepsilon$ by construction. From here, we can see that

$$\frac{\sum_{t=1}^{T_0} \delta^{t-1} x_i^{*t}}{\sum_{t=0}^{T_0-1} \delta^t} \geq \alpha_i \frac{\sum_{t=0}^{T-1} \delta^t}{\sum_{t=0}^{T_0-1} \delta^t} - \varepsilon \frac{\sum_{t=0}^{T-1-T_0} \delta^t}{\sum_{t=0}^{T_0-1} \delta^t} \geq (\alpha_i - \varepsilon) \frac{\sum_{t=0}^{T-1} \delta^t}{\sum_{t=0}^{T_0-1} \delta^t} \geq \alpha_i - \varepsilon$$

where the last two inequalities follow from the fact that $\delta^t > 0$ for all $t$ and that $T_0 \leq T$. It follows that

$$\sum_{t=1}^{T_0} \delta^{t-1} x_i^{*t} \geq (\alpha_i - \varepsilon) \sum_{t=0}^{T_0-1} \delta^t \quad (67)$$

$\square$

Now, we use this claim to evaluate the fairness guarantee of $\boldsymbol{x}'$. We claim that, over $T$ rounds, $\boldsymbol{x}'$ satisfies the fairness condition at a level $(1 - \varepsilon)(\alpha_i - \varepsilon)$ for $i \in \{1, 2\}$, i.e.,

$$\sum_{i=1}^T \delta^{t-1} x_i'^t \geq (1 - \varepsilon)(\alpha_i - \varepsilon) \sum_{t=0}^{T-1} \delta^t.$$

Since $x'^t_i = x^{*t}_i$ for $i \in [T_0]$, by Claim 5, it suffices to show

$$\sum_{t=T_0+1}^{T} \delta^{t-1} x'^t_i \geq (\alpha_i - \varepsilon) \sum_{t=T_0}^{T-1} \delta^t - \varepsilon(\alpha_i - \varepsilon) \sum_{t=0}^{T-1} \delta^t \tag{68}$$

Now, since $\delta \in (0,1)$ and $T \leq T_0$, we can observe that

$$\delta^{T_0} \frac{1 - \delta^T}{1 - \delta} \geq \frac{\delta^{T_0} - \delta^T}{1 - \delta}$$

Substituting $\delta_0^T = \varepsilon$ and recognizing these expressions as geometric series yields

$$\varepsilon \sum_{t=0}^{T-1} \delta^t \geq \sum_{t=T_0}^{T-1} \delta^t.$$

Now, noticing that $\alpha_i - \varepsilon \geq 0$ and rearranging, we see that

$$0 \geq (\alpha_i - \varepsilon) \sum_{t=T_0}^{T-1} \delta^t - \varepsilon(\alpha_i - \varepsilon) \sum_{t=0}^{T-1} \delta^t.$$

We reach the desired result of Equation (68) by noting that $\delta^{t-1} x^{*t}_i \geq 0$ for all $t$ and we have shown that, over $T$ rounds, $x'$ achieves a fairness level of at least $(1 - \varepsilon)(\alpha_i - \varepsilon)$.

Next, we consider the computational guarantee of $x'$. Since the fair allocation algorithm is recursive, its computational complexity increases exponentially in the number of rounds. In the approximation, we conduct exactly $T_0$ rounds of the fair allocation, therefore, the computational complexity can be bounded as follows:

$$\mathcal{O}\left(2^{T_0}\right) \leq \mathcal{O}\left(e^{T_0}\right) = \mathcal{O}\left(\varepsilon^{1/\log(\delta)}\right) = \mathcal{O}\left(\frac{1}{\varepsilon}^{\frac{1}{\log(1/\delta)}}\right)$$

Finally, we consider seller utility guarantees of $x'$. First, we note that, by Claim 5, the seller utility up to time $T_0$ under $x'$ is at least that of $x^*$ up to $T_0$ because the set of allocations over which $x'$ is optimal over the first $T_0$ rounds in terms of seller utility is a subset of those allocations over which $x^*$ is optimal. Second, since, under $x'$ we perform an unconstrained second-price auction in the remaining $T - T_0$ rounds, it follows that $x'$, must also guarantee a seller utility that is at least that of the optimal fair allocation $x^*$ over the second interval. Therefore, $x'$ guarantees a seller utility that is at least that of $x^*$ over all $T$ rounds. ∎

## A.8 Proof of proposition 4

First, we give the full proposition statement:

**Proposition 6.** *Suppose Assumptions 1-3 hold and that $\delta < 1$. Then, for any $\varepsilon \in (0, \min_i(\alpha_i)), \beta > 1 - \delta$, there exists an approximation to the optimal allocation $x''$ with the following properties:*

1. *$x''$ satisfies the fairness constraint over all rounds at a level of at least $(1-\varepsilon)(1-\beta)^2(\alpha_i-\varepsilon)$.*

2. *$x''$ guarantees that the sellers total utility is at least $(1 - \beta)$ of that of the optimal allocation.*

3. *$x''$ can be computed by calling the oracle $\mathcal{O}\left(\frac{1}{\varepsilon}^{\frac{1}{\beta}\log\left(\frac{1}{1-\delta}\right)+1}\right)$ times.*

Note that $\varepsilon, \beta$ can be chosen to achieve a given $c \leq \delta^2$ such that the proof statement in the body holds.

Now, we proceed with the proof by first defining $x''$ as follows: As in the approximation $x'$ presented in proposition 3, we assume under $x''$ that the fair auction runs over $T_0$ rounds (the same value $T_0$ which we detail later) where we partition these $T_0$ rounds into $T_0/\ell$ buckets of $\ell$ rounds (we later also define $\ell$ in detail) and approximate the discount factor as being constant within each bucket. Then, within each bucket, we recursively calculate the fair allocation at level $(1 - \beta)(1 - \alpha_i)$ with

respect to the approximated discontinuous discounting scheme. For the remaining $T - T_0$ rounds, we conduct a standard second-price auction.

In particular, as before, we take $T_0 := \frac{\log(\varepsilon)}{\log(\delta)}$ and we choose $\ell$ such that $\delta^\ell \simeq 1 - \beta$. Without loss of generality, we assume $T_0/\ell, \ell$ are integers. In the $k$th bucket, we approximate the discount factor of each round in the bucket with $\delta^{k-1}$ for $k \in [T_0/\ell]$.

Now, we characterize the allocation under $\boldsymbol{x}''$ in relation to the exact fair allocation $\boldsymbol{x}^*$ as defined in the proof of proposition 3 to show its fairness guarantee. First, observe the following relationship between the true discount factors and those of the discontinuous discounting scheme in $\boldsymbol{x}''$:

$$\sum_{t=0}^{T_0-1} \delta^t \geq (1-\beta) \sum_{k=1}^{T_0/\ell} \ell \cdot \delta^{\ell(k-1)}. \tag{69}$$

Therefore, by Claim 5 and our construction of the discontinuous discounting scheme, $\boldsymbol{x}^*$ satisfies the fairness constraint at the level $(1-\beta)(\alpha_i - \varepsilon)$ with respect to the discontinuous discounting scheme over the first $T_0$ rounds, i.e.,

$$\sum_{k=0}^{T_0/\ell-1} \delta^{\ell k} \sum_{t=k\ell+1}^{k\ell} x_i^{*t} \geq \sum_{t=1}^{T_0} \delta^{t-1} x_i^{*t} \geq (1-\beta)(\alpha_i - \varepsilon) \sum_{k=1}^{T_0/\ell} \ell \cdot \delta^{\ell(k-1)}.$$

From here, we characterize $\boldsymbol{x}''$ as the allocation such that, in the first $T_0$ rounds, we perform the fair allocation procedure that satisfies fairness constraint at a level $(1-\beta)(\alpha_i - \varepsilon)$ with respect to the discontinuous discounting scheme and a standard second-price auction in the remaining $T - T_0$ rounds. It follows that $\boldsymbol{x}''$ satisfies

$$\sum_{t=0}^{T_0-1} \delta^t x_i''^t \geq (1-\beta) \sum_{k=0}^{T_0/\ell-1} \delta^{\ell k} \sum_{t=k\ell+1}^{k\ell} x_i''^t \geq (1-\beta)^2(\alpha_i - \varepsilon) \sum_{t=0}^{T_0-1} \delta^t.$$

In other words, over the first $T_0$ rounds, $\boldsymbol{x}''$ satisfies the fairness constraint at a level of at least $(1-\beta)^2(\alpha_i - \varepsilon)$ with respect to the original discounting scheme. By the same manner as the proof of proposition 3, it follows that, over $T$ rounds, $\boldsymbol{x}''$ guarantees fairness at the level at least $(1-\varepsilon)(1-\beta)^2(\alpha_i - \varepsilon)$. Finally, by (69) and the same logic as the proof of proposition 3, we conclude that the seller's utility is at least $(1-\beta)$ that of $\boldsymbol{x}^*$ under $\boldsymbol{x}''$ over $T$ rounds. ∎

### A.9  Experiment Details

Here, we provide the full results of the experimentation summarized in the body of the paper.[6] For each combination of $\alpha_1, \alpha_2$ and each value of $T = 2, 3, 4$, we compute 10,000 iterations of the seller-optimal allocation satisfying the fairness constraint at level $\alpha_i$ for group $i \in [2]$ over 2 rounds with discount factor $\delta = 0.99$. Also note that buyer values are distributed as follows: $v_1^t \sim \text{Uniform}(0.5, 1.5)$, and $v_2^t \sim \text{Uniform}(0, 1)$ for $t \in [T], T = 2, 3, 4$. It is useful to note that, in the seller-optimal unconstrained mechanism, groups 1 and 2 are allocated the item with probabilities 0.69 and 0.31, respectively. Our implementation maintains all key assumptions outlined in the presentation of the mechanism. We report the difference in mean utility of the seller, group 1, and group 2 that satisfies the fairness constraint specified by a particular $\alpha_1, \alpha_2$ relative to the utility of the optimal unconstrained mechanism satisfying assumption 2 (i.e. the optimal fair mechanism at level $\alpha_1, \alpha_2 = 0$). Standard errors are reported for each value in Tables 1-6 and reflect the standard error of the mean which assumes sample independence.

---

[6]Note that additional experimental results are available in an extended version available at https://arxiv.org/abs/2406.00147

Table 1: Difference in seller utility relative to unconstrained optimal allocation with standard errors for $T = 2$

| $\alpha_2$ | $\alpha_1$ | 0.0 | 0.1 | 0.2 | 0.3 | 0.4 | 0.5 |
|---|---|---|---|---|---|---|---|
| 0.0 | | 0.0 | -0.0019 | -0.0015 | -0.004 | -0.0134 | -0.0624 |
| | | (0.0031) | (0.0032) | (0.0032) | (0.0032) | (0.003) | (0.0029) |
| 0.1 | | -0.2033 | -0.1957 | -0.2033 | -0.2016 | -0.2224 | -0.2679 |
| | | (0.0037) | (0.0037) | (0.0037) | (0.0037) | (0.0036) | (0.0034) |
| 0.2 | | -0.2688 | -0.267 | -0.2759 | -0.2715 | -0.2777 | -0.3291 |
| | | (0.0038) | (0.0038) | (0.0038) | (0.0038) | (0.0037) | (0.0036) |
| 0.3 | | -0.3435 | -0.344 | -0.3345 | -0.3416 | -0.3531 | -0.4061 |
| | | (0.0039) | (0.0039) | (0.004) | (0.0039) | (0.0038) | (0.0036) |
| 0.4 | | -0.4341 | -0.4285 | -0.4245 | -0.4227 | -0.4383 | -0.4828 |
| | | (0.0038) | (0.0039) | (0.0039) | (0.0039) | (0.0038) | (0.0036) |
| 0.5 | | -0.5297 | -0.5351 | -0.5337 | -0.5325 | -0.5394 | -0.587 |
| | | (0.0036) | (0.0036) | (0.0036) | (0.0036) | (0.0035) | (0.0033) |

Table 2: Difference in group 1 utility relative to unconstrained optimal allocation with standard errors for $T = 2$

| $\alpha_2$ | $\alpha_1$ | 0.0 | 0.1 | 0.2 | 0.3 | 0.4 | 0.5 |
|---|---|---|---|---|---|---|---|
| 0.0 | | 0.0 | 0.0065 | 0.0007 | 0.0019 | 0.023 | 0.0622 |
| | | (0.0036) | (0.0037) | (0.0037) | (0.0037) | (0.0037) | (0.0038) |
| 0.1 | | -0.0155 | -0.0058 | -0.0123 | -0.0081 | 0.0035 | 0.0587 |
| | | (0.0035) | (0.0036) | (0.0036) | (0.0035) | (0.0036) | (0.0037) |
| 0.2 | | -0.0232 | -0.0169 | -0.0186 | -0.0184 | -0.0018 | 0.0409 |
| | | (0.0034) | (0.0034) | (0.0034) | (0.0034) | (0.0035) | (0.0035) |
| 0.3 | | -0.0305 | -0.0317 | -0.0247 | -0.0267 | -0.0119 | 0.037 |
| | | (0.0033) | (0.0033) | (0.0033) | (0.0033) | (0.0034) | (0.0034) |
| 0.4 | | -0.0451 | -0.0413 | -0.0445 | -0.0387 | -0.0277 | 0.013 |
| | | (0.0032) | (0.0032) | (0.0032) | (0.0032) | (0.0032) | (0.0033) |
| 0.5 | | -0.0639 | -0.0609 | -0.0618 | -0.0598 | -0.0549 | 0.0002 |
| | | (0.0031) | (0.0031) | (0.0031) | (0.0031) | (0.0031) | (0.0032) |

Table 3: Difference in group 2 utility relative to unconstrained optimal allocation with standard errors for $T = 2$

| $\alpha_2$ | $\alpha_1$ | 0.0 | 0.1 | 0.2 | 0.3 | 0.4 | 0.5 |
|---|---|---|---|---|---|---|---|
| 0.0 | | 0.0 | -0.0021 | -0.0009 | -0.0037 | 0.0032 | 0.0226 |
| | | (0.0025) | (0.0026) | (0.0026) | (0.0025) | (0.0026) | (0.0027) |
| 0.1 | | 0.1773 | 0.1729 | 0.1789 | 0.1749 | 0.1807 | 0.1873 |
| | | (0.0033) | (0.0033) | (0.0033) | (0.0032) | (0.0033) | (0.0034) |
| 0.2 | | 0.2214 | 0.2187 | 0.2215 | 0.2194 | 0.2212 | 0.2337 |
| | | (0.0033) | (0.0034) | (0.0034) | (0.0033) | (0.0034) | (0.0035) |
| 0.3 | | 0.2634 | 0.2705 | 0.2626 | 0.2629 | 0.2742 | 0.2849 |
| | | (0.0034) | (0.0034) | (0.0034) | (0.0034) | (0.0034) | (0.0035) |
| 0.4 | | 0.3245 | 0.3228 | 0.3253 | 0.3205 | 0.3284 | 0.3424 |
| | | (0.0034) | (0.0035) | (0.0034) | (0.0035) | (0.0035) | (0.0036) |
| 0.5 | | 0.3923 | 0.3896 | 0.3933 | 0.3861 | 0.397 | 0.4075 |
| | | (0.0034) | (0.0035) | (0.0034) | (0.0034) | (0.0034) | (0.0035) |

Table 4: Difference in seller utility relative to unconstrained optimal allocation with standard errors for $T = 4$

| $\alpha_2$ | $\alpha_1$ | 0.0 | 0.1 | 0.2 | 0.3 | 0.4 | 0.5 |
|---|---|---|---|---|---|---|---|
| 0.0 | | 0.0 | -0.0006 | -0.0209 | -0.06 | -0.0642 | -0.1069 |
| | | (0.0035) | (0.0034) | (0.0035) | (0.0035) | (0.0035) | (0.0038) |
| 0.1 | | -0.2693 | -0.2578 | -0.2767 | -0.3308 | -0.324 | -0.3715 |
| | | (0.0038) | (0.0038) | (0.0038) | (0.0038) | (0.0038) | (0.0039) |
| 0.2 | | -0.4246 | -0.4145 | -0.4279 | -0.4767 | -0.4801 | -0.5395 |
| | | (0.0038) | (0.0037) | (0.0038) | (0.0038) | (0.0038) | (0.0038) |
| 0.3 | | -0.623 | -0.6251 | -0.6363 | -0.6913 | -0.6895 | -0.7614 |
| | | (0.0038) | (0.0037) | (0.0038) | (0.0038) | (0.0037) | (0.0037) |
| 0.4 | | -0.6836 | -0.6792 | -0.6955 | -0.7329 | -0.741 | -0.8184 |
| | | (0.0038) | (0.0038) | (0.0038) | (0.0038) | (0.0038) | (0.0036) |
| 0.5 | | -0.4898 | -0.4868 | -0.4955 | -0.5465 | -0.5498 | -0.6848 |
| | | (0.0036) | (0.0036) | (0.0036) | (0.0036) | (0.0035) | (0.0037) |

Table 5: Difference in group 1 utility relative to unconstrained optimal allocation with standard errors for $T = 4$

| $\alpha_2$ | $\alpha_1$ | 0.0 | 0.1 | 0.2 | 0.3 | 0.4 | 0.5 |
|---|---|---|---|---|---|---|---|
| 0.0 | | 0.0 | 0.0055 | 0.0117 | 0.0673 | 0.0652 | 0.1944 |
| | | (0.0054) | (0.0055) | (0.0055) | (0.0055) | (0.0054) | (0.0055) |
| 0.1 | | -0.0467 | -0.0412 | -0.035 | 0.0103 | 0.0164 | 0.1613 |
| | | (0.0053) | (0.0052) | (0.0053) | (0.0052) | (0.0053) | (0.0054) |
| 0.2 | | -0.0824 | -0.0812 | -0.0765 | -0.0269 | -0.0245 | 0.1163 |
| | | (0.0051) | (0.0051) | (0.0051) | (0.0051) | (0.0051) | (0.0052) |
| 0.3 | | -0.1372 | -0.1325 | -0.1204 | -0.0862 | -0.0712 | 0.0673 |
| | | (0.0049) | (0.0049) | (0.0049) | (0.005) | (0.0049) | (0.005) |
| 0.4 | | -0.1641 | -0.1592 | -0.1481 | -0.0988 | -0.0956 | 0.0514 |
| | | (0.0048) | (0.0048) | (0.0048) | (0.0048) | (0.0048) | (0.0049) |
| 0.5 | | -0.2634 | -0.2681 | -0.246 | -0.2055 | -0.1944 | -0.0228 |
| | | (0.0042) | (0.0042) | (0.0042) | (0.0042) | (0.0042) | (0.0043) |

Table 6: Difference in group 2 utility relative to unconstrained optimal allocation with standard errors for $T = 4$

| $\alpha_2$ | $\alpha_1$ | 0.0 | 0.1 | 0.2 | 0.3 | 0.4 | 0.5 |
|---|---|---|---|---|---|---|---|
| 0.0 | | 0.0 | -0.0058 | -0.0081 | -0.011 | -0.0087 | -0.0666 |
| | | (0.0043) | (0.0043) | (0.0043) | (0.0043) | (0.0043) | (0.0042) |
| 0.1 | | 0.2473 | 0.2397 | 0.2362 | 0.2397 | 0.2408 | 0.1908 |
| | | (0.0046) | (0.0046) | (0.0045) | (0.0045) | (0.0046) | (0.0045) |
| 0.2 | | 0.3692 | 0.3712 | 0.3722 | 0.3757 | 0.3814 | 0.3376 |
| | | (0.0046) | (0.0046) | (0.0046) | (0.0046) | (0.0046) | (0.0046) |
| 0.3 | | 0.5476 | 0.5475 | 0.5437 | 0.5486 | 0.5436 | 0.5274 |
| | | (0.0049) | (0.0048) | (0.0049) | (0.0049) | (0.0049) | (0.005) |
| 0.4 | | 0.5734 | 0.5748 | 0.5663 | 0.5591 | 0.5699 | 0.545 |
| | | (0.0049) | (0.005) | (0.005) | (0.005) | (0.0051) | (0.005) |
| 0.5 | | 0.2677 | 0.2783 | 0.2751 | 0.2692 | 0.2739 | 0.2945 |
| | | (0.0048) | (0.0048) | (0.0048) | (0.0048) | (0.0048) | (0.0048) |

All experiments were performed using hardware with the following specifications: Intel Xeon CPU @ 2.20GHz, 1 CPU core. The total experiment runtime was 23 minutes and 10 seconds and used 1.5 GB system RAM.

