# OpenReview forum: "Fair Allocation in Dynamic Mechanism Design"
_NeurIPS.cc/2024/Conference — NeurIPS 2024 poster_

### Official Review · Reviewer_ZeA8 · 2024-07-12

**Soundness:** 4
**Presentation:** 4
**Contribution:** 4
**Rating:** 7
**Confidence:** 3

**Summary:**

The paper explores an auction mechanism where an auctioneer aims to maximize discounted overall revenue while adhering to fairness constraints that ensure a minimum average allocation to two distinct groups. The study begins with a simple T = 1 scenario to establish the foundational optimal mechanism constraints, which include:

- Overall probability distribution to all buyers
- Preferential treatment for the group with an otherwise lower probability of winning

In a dynamic setting, the paper extends to explore recursive solutions that utilize price history to optimize revenue while maintaining fairness. This approach extends with incentives beyond the traditional second price auction optimality, adjusting for group-based discrimination in each round.

**Strengths:**

- The manuscript is exceptionally well-written, with all assumptions clearly justified and results thoroughly explained prior to their introduction. The logical flow and clarity of exposition make the results easily interpretable.
- Although brief, experiments using randomly generated datasets illustrate the practical application of the theoretical results, especially emphasizing the dynamic setting's theoretical contributions.
- The discussion on optimal allocation rules is insightful and presents a novel extension beyond existing literature, addressing new contexts of fairness in auction settings.

**Weaknesses:**

- The paper would benefit from a more comprehensive discussion of related works to better situate its contributions within the existing body of knowledge.
- Concluding remarks discussing potential future directions and the real-world applicability of the theory are missing. While the limited space of conference submissions is acknowledged, such discussion could significantly enhance the paper's impact.

**Questions:**

- Could the model be extended to more than two groups, and if so, would this add significant complexity or insight compared to the current settings?
- How standard is Assumption 1 used within the model, and how does it compare to assumptions typically made in similar studies?
- The term "dynamic" used to describe settings with T > 1 might need clarification or justification. Is there a more precise terminology that could better describe these recursive, history-dependent solutions? Dynamic usually refers to some flexibility in the allocation of items.

**Limitations:**

Addressed in the prior comments. Further experimentation on real world datasets could be useful but not a major limitation as this is largely a theoretical paper.

---

> ### Author Rebuttal · Authors · 2024-08-07
>
> We thank the reviewer for their valuable feedback and comments. Please find our answers below.
>
> > The paper would benefit from a more comprehensive discussion of related works to better situate its contributions within the existing body of knowledge.
>
> We acknowledge that the related work section should be expanded to include more papers. Due to space limitations, we were unable to include all the related papers in the submission version. We plan to provide a more comprehensive discussion of the related work in the main body or the appendix in the final version. To demonstrate that we already have the expanded related work available, we provide a brief summary of it here. We would also be happy to include any additional references that the reviewers suggest or find relevant.
>
> In particular, we plan to draw connections between our work and the literature that studies fair allocation **without** considering strategic agents, such as Procaccia and Wang [2014] , [Babaioff et al., 2022], [Budish, 2011], [Caragiannis et al., 2023], [Conitzer et al., 2017], and [Amanatidis et al., 2018]. These works fundamentally differ from our paper as there is no incentive involved, but we would ensure to include them in the updated related work version.
>
> Also, we would do a more broad discussion on papers that study fairness in the presence of strategic agents, including Lipton et al. [2004], Babichenko et al. [2023], Caragiannis et al. [2009], Sinha and Anastasopoulos [2015], Amanatidis et al. [2016], Amanatidis et al. [2017], Amanatidis et al. [2023b], Cole et al. [2013], and Babaioff and Feige [2022, 2024]. Note that these works differ from our work as they consider the static case and different notions of fairness, and some even consider social welfare rather than revenue. However, we plan to include all for the sake of completeness.
>
> > Concluding remarks discussing potential future directions and the real-world applicability of the theory are missing. While the limited space of conference submissions is acknowledged, such discussion could significantly enhance the paper's impact.
>
> We thank the reviewer for pointing out these shortcomings. Please see General Comment 1 on a number of real-world applications of our results. Also, we have discussed potential future directions in our response to reviewer WPVa. We will incorporate both in the final version of the paper using the additional one-page space.
>
> >Could the model be extended to more than two groups, and if so, would this add significant complexity or insight compared to the current settings?
>
> Yes, please see General Comment 2 in our global response.
>
> > How standard is Assumption 1 used within the model, and how does it compare to assumptions typically made in similar studies?
>
> Assumption 1, i.e., the regularity assumption, is quite common in the mechanism design literature, dating back to the seminal work of Myerson [1981], and has been used in many other mechanism design papers over decades. That said, the original Myerson [1981] paper introduces the ironing technique, which can be used to relax this assumption in their model. As we discuss in our response to Reviewer ssTM, this technique can be applied in our setting as well to extend our results to cases where Assumption 1 does not hold. We have provided the proof in our response to Reviewer ssTM and will include it in the final version of the paper.
>
> > The term "dynamic" used to describe settings with $T > 1$ might need clarification or justification. Is there a more precise terminology that could better describe these recursive, history-dependent solutions? Dynamic usually refers to some flexibility in the allocation of items.
>
> The choice of the term “dynamic” primarily follows the mechanism design literature, as “dynamic mechanism design” is the customary terminology used to refer to scenarios in which items are allocated over time through incentive-aware mechanisms (see reference [2] in our submission for a detailed reference).

---

> > ### Comment · Reviewer_ZeA8 · 2024-08-07
> >
> > Thanks for such a detailed response! This clarifies my noted questions and hope the authors revise the text accordingly to clarify these points for other readers. I'm keeping my score a 7 as I believe the paper should be accepted and would be a nice inclusion to the upcoming conference.

---

### Official Review · Reviewer_ssTM · 2024-07-13

**Soundness:** 4
**Presentation:** 4
**Contribution:** 3
**Rating:** 7
**Confidence:** 4

**Summary:**

This paper studies the incorporation of fairness constraints into revenue-optimal single-item auctions. Specifically, it focuses on a scenario with two groups bidders. Within each group, bidders' private valuations are sampled i.i.d. from a distribution, with the two groups having different valuation distributions. The fairness constraint is defined by two numbers, $\alpha_1$ and $\alpha_2$, requiring the mechanism to ensure that the expected allocation to group $i$ is at least $\alpha_i$. The objective is to identify the revenue-optimal EPIC and IR mechanism that meets this fairness criterion.


The submission first investigates the static case where only one round of the auction is conducted and characterizes the optimal mechanism. The authors demonstrate that, compared to the optimal auction without fairness constraints, the optimal mechanism with fairness constraints subsidizes all bidders in the virtual value space to enhance their chances of allocation. Additionally, the paper observes that the optimal mechanism provides subsidies to the disadvantaged group. From a technical perspective, the argument extends Myerson’s original proof of optimal auctions without fairness considerations in a relatively straightforward manner.

Next, the authors extend their analysis to a dynamic auction setting where the auction is conducted over $T$ rounds, with an item being auctioned in each round. Valuations can vary across rounds. In this dynamic setting, the fairness constraint imposes a lower bound on the expected number of allocated items in future rounds based on the allocation history of previous rounds. Bidder incentives become more complex, as they might underbid in the current round to gain an advantage in future rounds with higher-valued items due to the fairness constraints. The authors characterize the optimal mechanism using backward induction, noting that the optimal mechanism starting from round $t$ depends only on the remaining fairness quota for each group. With this insight, the optimal mechanism starting from round $t$ can be determined by implementing Myerson’s auction after adjusting the bidder's value by the externality (the decrease in future revenue from awarding the item to the bidder in the current round). With a discounting factor, finding the exact optimal mechanism requires exponential time. The authors propose methods to efficiently approximate it through early stopping and discretization.

The paper concludes with a numerical experiment illustrating how different fairness constraints impact revenue and bidders’ utilities in each group.

Minor comments:
- There is an extra symbol in equation (6).
- It would be much clearer to add a description of the allocation for each region in Figure 1.

**Strengths:**

- The problem is well-motivated. I believe studying fairness notions in dynamic auction settings has its potential. In addition, the specific model considered in the paper feels reasonable.
- The paper fully characterizes the optimal mechanism and provides some high-level interpretations.
- The paper is well-written. Technical parts are easy to follow.

**Weaknesses:**

- Experiments seem too simple, it would be better if the authors conducted a more extensive experiment.
- Though the authors claim that results should extend to more than 2 groups, it would better to include some formal statements.

**Questions:**

- Do the results still hold when bidders in one group have different valuation distributions?
- The paper assumes that all distributions are regular. Do main results extend to arbitrary distribution by ironing?

**Limitations:**

Yes, the authors have adequately addressed the limitations.

---

> ### Author Rebuttal · Authors · 2024-08-07
>
> We thank the reviewer for their valuable feedback and comments. Please find our answers below.
>
> > Minor Comments
>
> We thank the reviewer for pointing out the typo. We also agree with the suggested improvements to our figures and would incorporate them in the final version.
>
> > Experiments seem too simple, it would be better if the authors conducted a more extensive experiment.
>
> We acknowledge the simplicity of our initially included experimentation; we agree that extensions that consider complexities including those resulting from increasing the number of rounds and buyers or considering other value distributions, for instance, will be valuable to illustrate characteristics of the optimal fair mechanism under a variety of conditions. We intend to include a broader set of experimental findings from our ongoing extensions. That said, we have already run a number of additional experiments, as illustrated in the General Comment 3 in our global response.
>
> > Though the authors claim that results should extend to more than 2 groups, it would be better to include some formal statements.
>
> We thank the reviewer for their suggestion. Please see General Comment 2 in our global response.
>
> > Do the results still hold when bidders in one group have different valuation distributions?
>
> We thank the reviewer for this intriguing question. Yes, our analysis extends to that case as well. We made this assumption primarily based on our motivating examples (see General Comment 1, for instance). Specifically, we are looking for group-fairness guarantees, where we have homogeneous buyers within each group, but the distribution of values differs across groups. For instance, this could be because buyers in one group have lower incomes and, hence, lower willingness to pay.
>
> That said, as we stated above, our analysis extends to the case where we have heterogeneous buyers within each group. The main difference in this case is that, within each group, the buyer with the **highest virtual value** has the chance of winning the item, rather than the buyer with the highest value (which is the current result). Furthermore, we should compare the highest virtual values of the two groups to decide on allocation. Let us formalize this for the static case; the dynamic case follows similarly.
>
> Notice that in this case, the virtual value of buyer $(i,k)$ is denoted by $\phi_{i,k}(v_{i,k})$. Now, part (i) of Theorem 1 would change to: "If the item is allocated to group $i$, then it is allocated to the buyer in group $i$ with the **highest virtual value**." Regarding part (ii), equations (7a) and (7b) would hold, with the difference that we have to define $\phi_i(v_i) := \max_{i,k} \phi_{i,k}(v_{i,k})$. The proof of this updated result would follow identically to the proof of the current result.
>
> > The paper assumes that all distributions are regular. Do main results extend to arbitrary distribution by ironing?
>
> We thank the reviewer for pointing out this relevant extension. Yes, the ironing technique can be used to extend our result to arbitrary distributions. We provide a brief summary of the result here, and we will include a detailed result in the appendix in the final version, along with a discussion in the main body. To simplify the notation here, we suppress the dependence on time $t$.
>
> Let $h_i(.)$ be the virtual value function in the quantile space, i.e., $h_i(q) = \phi_i(F_i^{-1}(q))$ for any $q \in [0,1]$. Also, let $H_i$ be its cumulative virtual value function, i.e, $H_i(q) = \int_0^q h_i(q') dq'$. Now, let us recall the ironing technique from Myerson [1981]. We define $G_i: [0,1] \to \mathbb{R}$ as the convex hull of $H_i$, i.e., the largest convex function underestimator of $H_i$, and denote its derivative by $g'_i(\cdot)$. Now, the ironed virtual value function $\tilde{\phi}_i(\cdot)$ is simply defined as $\tilde{\phi}_i(v) = g_i(F_i(v))$. Notice that, given that we have dropped Assumption 1, $\phi_i(\cdot)$ is not necessarily monotone. However, given the ironing procedure, $\tilde{\phi}_i(\cdot)$ is monotone. In fact, when Assumption 1 holds, i.e., for regular distributions, $H(\cdot)$ is convex itself, and hence $g_i = h_i$ which implies $\tilde{\phi}_i = \phi_i$.
>
> Now, as Myerson [1981] establishes, the seller's revenue can be cast as
> $$\mathbb{E}\_{\pmb{v}} \left [ \sum_{i \in [2],k \in [n]} \phi_{i}(v_{i,k}) x_{i,k}(\pmb{v}) \right] =
> \mathcal{R} - \mathcal{E}
> $$
> with
> $$
> \mathcal{R} := \mathbb{E}\_{\pmb{v}} \left [ \sum_{i \in [2],k \in [n]} \tilde{\phi}\_{i}(v_{i,k}) x_{i,k}(\pmb{v}) \right]$$
> and
> $$
> \mathcal{E} := \sum\_{i \in [2],k \in [n]} \mathbb{E}\_{\pmb{v}\_{-(i,k)}} \left [ \int\_{\underline{v}\_i}^{\bar{v}\_i} (H_i(F_i(v)) - G_i(F_i(v))) ~ dx_{i,k}(v, \pmb{v}_{-(i,k)}) \right ].
> $$
>
> Now, notice that since $\tilde{\phi}\_i$ is monotone, our current result gives us the allocation that maximizes $\mathcal{R}$ subject to the fairness constraint, which, as Theorems 1 and 2 suggest, is in the form of $\max_{k} \tilde{\phi}\_1(v_{1,k}) - \max_{k} \tilde{\phi}\_2(v_{2,k}) \lesseqgtr \gamma$ for some $\gamma$.
>
> Therefore, it suffices to show that for allocations in this form, $\mathcal{E}$ is zero. Notice that $\mathcal{E}$ can only be positive when $H_i(F_i(v)) > G_i(F_i(v))$ and $dx\_{i,k}(v, \pmb{v}\_{-(i,k)}) > 0$. However, by the definition of the convex hull, when $H_i(F_i(v)) > G_i(F_i(v))$, then $G_i$ is linear in a neighborhood of $v$, which means $g'_i(v) = 0$, implying that $\tilde{\phi}\_i(v)$ is constant in a neighborhood of $v$. Now, as the value of others is fixed and the virtual value of buyer $(i,k)$ is also constant in a neighborhood of $v$, given the above form of the allocation, $x\_{i,k}(v, \pmb{v}\_{-(i,k)})$ also remains constant in that neighborhood, which implies $dx\_{i,k}(v, \pmb{v}\_{-(i,k)}) = 0$. This completes the proof that $\mathcal{E} = 0$.

---

> > ### Comment · Reviewer_ssTM · 2024-08-14
> >
> > Thank you for the detailed response. I'm pleased with the proposed clarifications and modifications, so I'll be maintaining my score.

---

### Official Review · Reviewer_WPVa · 2024-07-13

**Soundness:** 3
**Presentation:** 3
**Contribution:** 3
**Rating:** 6
**Confidence:** 2

**Summary:**

This paper studies a fair allocation problem where an auctioneer sells an indivisible good to two groups of buyers every round for T rounds. The auctioneer’s objective is to maximize their discounted revenue with fairness constraints. The authors show that for the static case with T=1, the optimal mechanism subsidizes one group to meet the fairness constraints, and may increase the probability of allocating the item to both groups by reducing the reserve price (compared to Myerson’s auction). For the dynamic case with multiple rounds, they characterize the optimal allocation by a set of recursive functions. They establish that in the optimal allocation, to incentivize truthful value reporting, the seller pays a participation reward for the winning group, but also charges the buyers an entry fee. Similar to the static case, the optimal allocation involves subsidizing in favor of one group. Finally, they present an approximation algorithm to solve the recursive equations.

**Strengths:**

The problem of allocating an indivisible good with some fairness constraints is well motivated. This paper extends previous work with a similar fairness definition in the single group and single round setting to two groups and multiple rounds, and characterizes different types of subsidization for this setting. This paper is mostly well written and related work is adequately cited.

**Weaknesses:**

* Minor comments
  * The discount factor $\delta$ is mentioned in the introduction, better to make sure it’s also defined in the Model section for notation reference.
  * Page 1, line 11: “on one hand” -> “on the one hand”
  * Page 6, line 231: “methods is provided” -> “methods are provided”
  * Page 6, line 253: “Aggregated-SP seem to be” -> “Aggregated-SP seems to be”
  * Page 8, line 315: “provide upper bound” -> “provide an upper bound”

**Questions:**

* Would you add some motivation for the discount factor?
* Would you add some discussions of the limitations of this work and future directions in the main paper?

**Limitations:**

The authors discussed limitations in the paper checklist form, but those are not explicitly mentioned in the main paper, potentially because of page limitations.

---

> ### Author Rebuttal · Authors · 2024-08-07
>
> We thank the reviewer for their valuable feedback and comments. Please find our answers below.
>
> > The discount factor  is mentioned in the introduction, better to make sure it’s also defined in the Model section for notation reference.
>
> We thank the reviewer for this suggestion and acknowledge that the discount factor should also be defined with the model. We will address this in the final version.
>
> > Minor Comments and Typos
>
> We thank the reviewer for pointing out all the typos, and we would fix them in the final version.
>
> > Would you add some motivation for the discount factor?
>
> The discount factor is commonly used in settings where we allocate items over time to account for the time value of money, i.e., one dollar today is worth more than one dollar in a year. In the context of studying fairness, it is also important as it ensures that items are not disproportionately allocated to one group initially, but are distributed reasonably over time.
>
> That said, it is worth noting that all our results extend to the case where we use a discount factor $\delta = 1$, i.e., the overall utility of each buyer is the average of utilities over $T$ rounds. The results for the dynamic setting would hold in this case, and Fact 3 elaborates that we can compute the optimal allocation efficiently under these conditions.
>
> > Would you add some discussions of the limitations of this work and future directions in the main paper?
>
> We thank the reviewer for motivating us to further discuss the limitations of our work and potential future directions. We will add these discussions using the additional one page allowed for the main body in the final version.
>
> Regarding limitations, our paper makes a number of theoretical assumptions, including the independence of buyers' values and the regularity of the distribution values. As we discuss in our response to Reviewer ssTM, the regularity assumption can be relaxed using the ironing technique, but studying the case with interdependent values requires more work and could be the topic of future research. Additionally, our paper assumes that the utility of each user is bilinear in the allocation and their value. Extending our results to a more general class of utility functions could be another potential direction for future work.

---

> > ### Comment · Reviewer_WPVa · 2024-08-12
> >
> > I thank the authors for their response. I will keep my score.

---

### Official Review · Reviewer_5sNL · 2024-07-14

**Soundness:** 3
**Presentation:** 2
**Contribution:** 3
**Rating:** 5
**Confidence:** 2

**Summary:**

The authors study the problem of dynamic mechanism design when in which for $T$ rounds an auctioneer sells an indivisible good to two groups of people. The goal is to design a mechanism which incentivizes the agents first to participate in the auction and second to bid truthfully and moreover maximizes the discounted revenue of the seller while guaranteeing a minimum expected allocation for each group. The proposed mechanism utilizes two types of subsidization. One is to reduce the reserved bid to increase the probability of allocation for all buyers and another is to favor the group which otherwise is less probable to win the good. These hold even when $T=1$. For $T>1$, the seller rewards the winner to incentivize truth-telling since otherwise, the agents might benefit from underbidding to increase the probability of winning in later rounds (which they might expect to value more). Moreover, the seller also charges the participants with an entry fee which is the expected reward payment they lose by not participating.

The proposed mechanism finds the optimal allocation in exponential time in terms of $T$. However, the authors propose an efficient approximation scheme and a poly-time constant approximation scheme. They also implement their mechanism and compare the utility to the case with no fairness constraints.

**Strengths:**

The studied problem of achieving fairness in mechanism design where agents are strategic is important and interesting. While the paper is quite notation-heavy, intuitions are provided to better understand what is going on.

**Weaknesses:**

The paper is not clear in some parts. The parameter $\delta$ is never clearly defined and it was very confusing to see it in line 112 without any proper previous description. Furthermore, the setting is not motivated. It would be useful to mention some real-world scenarios in which these groups are formed and the goal is to be fair towards the groups as a whole while allocating the goods to individuals. Also, the fairness is defined as giving each group a certain amount of goods in expectation. It is not intuitive at all why this is a good fairness measure while the value of the agents for the goods are totally ignored in this notion. Also, it would be very useful to have a theorem in the paper which is stand-alone and concisely mentions the main contribution of the paper.

**Questions:**

Could you please clarify the points mentioned in the last section. Namely,
1. please motivate the studied setting by real-world instances.
2. please justify the proposed fairness notion.

**Limitations:**

The limitations are addressed in a sense that the assumptions on the studied setting are mentioned.

---

> ### Author Rebuttal · Authors · 2024-08-07
>
> We thank the reviewer for their valuable feedback and comments. Please find our answers below.
>
> > The parameter $\delta$ is never clearly defined and it was very confusing to see it in line 112 without any proper previous description.
>
> We defined the discount factor $\delta$ earlier in the introduction (see line 34). However, we acknowledge that it should also be defined in the modeling section and we apologize for the lack of clarity there. We will address this in the final version.
>
> > It would be useful to mention some real-world scenarios in which these groups are formed and the goal is to be fair towards the groups as a whole while allocating the goods to individuals. Also, the fairness is defined as giving each group a certain amount of goods in expectation. It is not intuitive at all why this is a good fairness measure while the value of the agents for the goods are totally ignored in this notion.
>
> We thank the reviewer for motivating us to further elaborate on the application of our results. Please see General Comment 1 regarding a number of examples in this regard.
>
> We would like to use the first example on housing allocations to address the second part of your question. It is important to note that a buyer’s value is not always based solely on the intrinsic value of the item; it often reflects the buyer’s willingness to pay. In the housing example, a house in a good location might have a high value for both low-income and high-income groups, but their willingness to pay is limited by their ability to afford it. In such scenarios, the allocation ratio is a more effective approach to mitigate allocation inequality compared to using buyers’ value. This is why most regulations regarding fair housing allocation focus on the percentage of housing allocated to low-income groups as a measure of success.
>
> We hope these examples help to clarify and further motivate the formulation and model we study in this paper. We would also include this discussion in the final version of the paper.
>
> > Also, it would be very useful to have a theorem in the paper which is stand-alone and concisely mentions the main contribution of the paper.
>
> We see our paper as providing a framework for studying the fair allocation of goods through dynamic mechanism design. Our paper has three main contributions, which build on each other to complete the story. The first result, Theorem 1, illustrates the optimal allocation in the static case where we run the auction for one round. This result not only provides intuition on how fairness requirements change the optimal allocation but also serves as a basis for the dynamic case. The second main result, Theorem 2, demonstrates how we can find the optimal allocation recursively in the dynamic case. This can be seen as the main contribution of the paper. Finally, Propositions 3 and 4 aim to provide a computationally tractable framework for finding an approximation of the optimal allocation outlined by the previous theorem, making the result more accessible for applications. We hope this brief roadmap clarifies the main contributions of our paper.

---

> ### Comment · Reviewer_5sNL · 2024-08-09
>
> I sincerely thank the authors for their thorough response. With the provided examples and explanation, I understand the significance of the work and the motivation behind the proposed fairness criteria better. I increased my score.

---

### Author Rebuttal · Authors · 2024-08-07

**Global Responses:**

We thank the review team for their thoughtful and detailed comments. Here, we provide our general answers before addressing each reviewer's questions separately.

**General comment 1: Motivating examples based on real-world applications**

Auctions are used in various real-world applications, and in many of them, fairness considerations play an important role. In what follows, we provide a number of examples of such applications:

First, auctions are commonly used for allocating houses, either by private entities or government agencies. At the same time, several policies in different countries address the housing needs of low- and moderate-income households, from Affordable Housing Quotas in the United Kingdom to tax credits and vouchers in the United States. Our framework can be seen as an approach to combine such fairness considerations with auctions frequently used for allocating new houses.

Second, the federal government in the United States uses auctions to choose contractors for certain types of procurements. The United States Small Business Act sets targets for contracting with specific categories of businesses (e.g., woman-owned businesses, veteran-owned businesses, historically underutilized zones). These are yearly targets, and our approach would allow for these targets to be met as efficiently as possible.

Third, governments use auctions to decide on telecommunications licenses, i.e., which band of the electromagnetic spectrum should be used by each company to transmit signals. Ensuring smaller or regional companies have a chance to compete with larger national companies is one of the considerations in such allocations, which aligns with our framework. In fact, the Communications Act of 1934’s “equal opportunity” section requires that radio and television broadcasters provide “reasonable access” to all major political candidates. There has also been a recent push for the same requirement for digital/social media ads, which are sold algorithmically via auction.

Last but not least, auctions are used in environmental settings, including fishing rights, and there has been discussion about using auctions for the allocation of water rights given the growing conflicts among states and countries regarding access to water resources. It is evident that having a framework that motivates the fair allocation of resources among different agents while running auctions over time can be very applicable in these settings as well.


**General comment 2: Extension to more than two groups**


Regarding the question raised by two of the reviewers, we would like to briefly discuss how our results and insights extend to the case where we have more than two groups. Due to the character limit, we include the static case here.

In particular, let us see how Theorem 1 changes when we have $L$ groups. Part (i) remains unchanged, meaning that if the item is allocated to group $i$, it is allocated to the buyer with the highest value (and let’s denote this highest value by $v_i$). Then, we have the following:

*There exist nonnegative numbers $\eta_1, \cdots, \eta_L$ that characterize the optimal allocation in the following way: the item is allocated to group $i$ if and only if $\phi_i(v_i) + \eta_i \geq 0$ and $\phi_i(v_i) + \eta_i \geq \phi_j(v_j) + \eta_j$ for any $j \neq i$.*

Notice that the insight behind this result is very similar to the current Theorem 1 for two groups. $\eta_i$ is the subsidy to group $i$ and $\min_i \eta_i$ can be seen as the overall subsidy to the society (similar to $\gamma$ in our current draft).

The proof involves several steps, but the idea is similar to the case with two groups: we show that the best way to help groups that do not get the item with sufficient probability in the unconstrained case is to shift the allocation boundary of the unconstrained case towards them. More formally, here are the steps of the proof:

First, let $x^*$ denote the optimal allocation. If $x^*$ is not in the above form, we will show that through a series of steps, it can be transformed into an allocation of that form. Furthermore, we will establish that in this process, the seller's revenue can only increase, and the fairness constraints are not violated. While the intermediate allocations are not necessarily monotone, the final result will be a monotone (and hence a legitimate) allocation. In this process, we make use of the following lemma:

*Lemma 1: For any $i$ and $j$, there exists $\eta_{i,j}$ such that, if the item is allocated to one of the groups $i$ or $j$, then it is allocated to group $i$ if and only if $\phi_i(v_i) - \phi_j(v_j) \geq \eta_{i,j}$.*

If there are multiple choices for $\eta_{i,j}$, we pick the one with the smallest absolute value. Now, it remains to show that the $\eta_{i,j}$'s are indeed connected to each other. To do so, we show the following lemma:

*Lemma 2: Suppose $\eta_{i,j} \geq 0$ and $\eta_{j,k} \geq 0$. Then, we have $\eta_{i,k} = \eta_{i,j} + \eta_{j,k}$.*

The proof of this lemma is very similar to the way we prove $\eta_2 = \eta_1 + \gamma$ in Appendix A.1 of the submission draft (starting from line 491).

We hope this result establishes how our insight carries over to multi-group fairness. We will include a detailed result for both the static and dynamic cases as a new appendix in the final version of our paper.


**General comment 3: Additional experiments**


In response to reviewers' comments, we have conducted a number of additional numerical experiments (please see the attached file). In particular, we have included experiments that extend our initial utility comparison for a larger number of rounds, $T$. We have also included experiments comparing our fair mechanism to one that achieves fairness via set-asides, complementing the existing literature on their comparative merits (e.g., Athey et al. [2013], Pai and Vohra [2012]).

---

### Decision · Program_Chairs · 2024-09-25

**Decision:**

Accept (poster)

**Comment:**

Reviewers found the problem well-motivated, the characterizations and the algorithm interesting, and the paper well-written. Review also appreciated the clear logic flow and high-level proof ideas and the experiments. No significant concerns remain after the rebuttal. All reviews agreed that this is a nice piece of work for NeurIPS. We hope the authors find the reviews helpful. Thanks for submitting to NeurIPS!